# The causal role of the somatosensory cortex in prosocial behaviour

Selene Gallo[1], Riccardo Paracampo[1,2,3], Laura Müller-Pinzler[1,4],
Mario Carlo Severo[1‡], Laila Blömer[1†], Carolina Fernandes-Henriques[1†],
Anna Henschel[1†], Balint Kalista Lammes[1†], Tatjana Maskaljunas[1†], Judith Suttrup[1],
Alessio Avenanti[2,3], Christian Keysers[1,5], Valeria Gazzola[1,5]*

[1]Netherlands Institute for Neuroscience, Royal Netherlands Academy of Art and Sciences (KNAW), Amsterdam, Netherlands; [2]Department of Psychology, Center for Studies and Research in Cognitive Neuroscience, University of Bologna, Cesena, Italy; [3]IRCCS Fondazione Santa Lucia, Rome, Italy; [4]Department of Psychiatry and Psychotherapy, Social Neuroscience Lab, University of Lübeck, Lübeck, Germany; [5]Faculty of Social and Behavioural Sciences, University of Amsterdam, Amsterdam, Netherlands

*For correspondence:
v.gazzola@nin.knaw.nl

†These authors contributed equally to this work

Present address: ‡Department of Experimental Clinical and Health Psychology, Ghent University, Ghent, Belgium

Competing interests: The authors declare that no competing interests exist.

**Abstract** Witnessing another person's suffering elicits vicarious brain activity in areas that are active when we ourselves are in pain. Whether this activity influences prosocial behavior remains the subject of debate. Here participants witnessed a confederate express pain through a reaction of the swatted hand or through a facial expression, and could decide to reduce that pain by donating money. Participants donate more money on trials in which the confederate expressed more pain. Electroencephalography shows that activity of the somatosensory cortex I (SI) hand region explains variance in donation. Transcranial magnetic stimulation (TMS) shows that altering this activity interferes with the pain–donation coupling only when pain is expressed by the hand. High-definition transcranial direct current stimulation (HD-tDCS) shows that altering SI activity also interferes with pain perception. These experiments show that vicarious somatosensory activations contribute to prosocial decision-making and suggest that they do so by helping to transform observed reactions of affected body-parts into accurate perceptions of pain that are necessary for decision-making.
DOI: https://doi.org/10.7554/eLife.32740.001

## Introduction

Prosocial behavior — actions intended to benefit others despite costs to self (*Batson, 1981*) — is important in social animals but poorly understood. The role of empathy in motivating prosocial behavior is intuitive but also intensely debated (*Bloom, 2017*; *Zaki, 2017*). Some researchers have shown that people are more likely to engage in prosocial behaviour when they feel empathy for the person in distress (*Batson, 1981*) and that self-reported emphatic concern is related to pro-social behaviour (*FeldmanHall et al., 2015*). Others have shown that empathy is a poor predictor of prosociality (*Vachon et al., 2014*; *Jordan et al., 2016*), with prosocial decisions often driven by other motives (e.g. status; *Bloom, 2017*). Unfortunately, experiments that specifically manipulate brain activity in empathy-related regions and that measure prosociality are missing, limiting our neuroscientific understanding of whether and how empathy is mechanistically linked to prosociality (*Keysers and Gazzola, 2017*; *Zaki et al., 2016*). Here, we use a combination of electroencephalography (EEG), transcranial magnetic stimulation (TMS) and transcranial direct current stimulation (tDCS) to explore whether altering somatosensory activity that vicariously represents the pain of

**eLife digest** When we experience physical pain, certain areas in our brain that process bodily sensation and emotions switch on. If we see someone else in pain, many of the same regions also get activated. In contrast, convicted criminals with psychopathic traits have less activation in these areas of the brain when witnessing someone's pain; they also show less empathy and disregard the needs of others. This suggests that a lack of this 'shared activations' may lead to problems in empathy. In fact, many scientists believe that shared activations are why we feel empathy for people in pain, and why we are driven to help them. Yet, there is little direct evidence about how the activity in the pain processing parts of the brain actually influences helpful behavior. As a result, some scientists now argue that empathy-related processes may actually contribute very little to helping behavior.

Gallo et al. designed an experiment where participants watched videos of someone having their hand swatted with a belt, and showing different levels of pain as a result. The volunteers could decide to reduce the amount of pain the person received by donating money they could have taken home. The more pain the participants thought the victim was in, the more money they gave up to lessen it. During the study, the activity in the brain region that processes pain in the hand was also measured in the participants. The more active this region was, the more money people donated to help.

Then, Gallo et al. used techniques that interfered with the activity of the brain area involved in perceiving sensations from the hand. This interference changed how accurately participants assessed the victim's pain. It also disrupted the link between donations and the victim's perceived pain: the amount of money people gave no longer matched the level of pain they had witnessed. This suggests that the brain areas that perceive sensations of pain in the self, which evolved primarily to experience our own sensations, also have a social function. They transform the sight of bodily harm into an accurate feeling for how much pain the victim experiences. The findings also show that we need this feeling so we can adapt our help to the needs of others. In the current debate about the role of empathy in helping behaviors, this study demonstrates that empathy-related brain activity indeed promotes helping by allowing us to detect those that need our assistance.

Understanding the relationship between helping behavior and the activity of the brain may further lead to treatments for individuals with antisocial behavior and for children with callous and unemotional traits, a disorder that is associated with a lack of empathy and a general disregard for others.

DOI: https://doi.org/10.7554/eLife.32740.002

others would alter the decision to donate money to alleviate that pain, and whether alterations in pain perception mediate this effect.

Witnessing somebody in pain activates two networks, depending on the nature of the stimulus (*Keysers et al., 2010*; *Lamm et al., 2011*). If the pain of the other is deduced from abstract symbols or facial expressions alone, a network involving the anterior insula (AI) and the anterior cingulate cortex (ACC) is activated. The AI and ACC activity correlates with personal distress (*Singer et al., 2004*), perceived unpleasantness (*Rainville et al., 1997*) and perceived pain intensity (*Lamm et al., 2011*), and is therefore thought to code for the unpleasantness of the pain of the other (*Lamm et al., 2011*). This network is often referred to as the affective path. If the injured body part is visible and in the focus of attention, the somatosensory cortices (SI and SII) are also recruited (*Bufalari et al., 2007*; *Cheng et al., 2008*; *Keysers and Gazzola, 2009*; *Lamm et al., 2011*; *Nummenmaa et al., 2012*; *Morrison et al., 2013*; *Ashar et al., 2017*; *Christov-Moore and Iacoboni, 2016*) and consequently included in the network of regions participating in human empathy (*Keysers et al., 2010*; *de Waal and Preston, 2017*).

The affective and somatosensory networks are also active when experiencing pain (*Melzack, 2001*; *Iannetti and Mouraux, 2010*; *Lockwood et al., 2013*). Because of that, many interpret vicarious activation while witnessing the pain of others as the neural correlate of emotional contagion, feeling vicariously what we see someone else experience, a core component of empathy (*Koban et al., 2013*; *Corradi-Dell'Acqua et al., 2011*; *Jackson et al., 2006*; *Lamm et al., 2011*;

*Cui et al., 2015*; *Singer et al., 2004*). In this perspective, the emotional states of others are understood through personal, embodied representations that allow empathy and accuracy in perceiving other emotions to increase on the basis of the observer's past experiences (*Preston and de Waal, 2002*; *de Waal and Preston, 2017*). Pharmacological studies provide causal evidence for the involvement of our own experience of pain, via the insula and anterior cingulate cortex, in perceiving the pain of other individuals (*Rütgen et al., 2015, 2017*). This evidence pin-points to the opioidergic circuit, which might code representations of aversive outcomes that are derived through both direct and indirect experiences (*Haaker et al., 2017*). Much like the original proposal of Adam Smith (*Smith, 1759*), the pain vicariously felt because of the recruitment of these pain-related regions while viewing the pain of others is then thought to motivate prosocial behavior, which then simply serves to reduce the vicariously felt pain (*Batson, 1981*; *Hein et al., 2010*, *2016*; *Ma et al., 2011*; *Tomova et al., 2016*).

Testing the causal contribution of mapping the pain of others onto our own pain representations in social decision-making has remained poorly explored in neuroscience because most scientists focus on the affective network (AI, ACC), which lies too deep in the brain to allow its selective targeting with traditional non-invasive neuro-manipulation tools (*Keysers and Gazzola, 2017*). Here, we leverage the less investigated hand representation of SI, which is superficial and reachable with TMS, to address two questions: does activity in the somatosensory cortex (measured using EEG) explain prosocial behavior on a trial-by-trial basis and does disturbing (with TMS) this activity alter decision-making? We explore prosocial behavior using a costly helping paradigm (*Figure 1A*), in which participants make a moral decision between two conflicting motives: maximizing their financial gains and minimizing the pain of another (see *FeldmanHall et al. [2015]*) for use of a similar tradeoff scenario). We then measure and alter brain activity in the hand region of SI to explore the impact of this activity on the decision-making.

If somatosensory activity contributes to our decision to help, does it do so by being necessary for an accurate perception of how much pain other people experience? Whether somatosensory activation levels reflect the intensity of the pain experienced by others remains unclear (*Lamm et al., 2011*; *Morrison et al., 2013*), and we therefore use data from a third experiment in which high-definition tDCS (HD-tDCS) is used to alter SI activity to measure whether an SI perturbation also alters the accuracy of pain perception.

As mentioned above, somatosensory cortices are mainly involved when the injured body parts are visible. In addition, ventral regions of SI have been reported to be involved in emotional facial perception (e.g *Adolphs et al., 2000*; *van der Gaag et al., 2007*; *Preston and de Waal, 2002*; but see *Rütgen et al., 2015* for absence of SI activity in the perception of facial expression of pain). Given its somatotopic organization, SI activity could therefore be involved in pain perception in two ways. If we only see the painful reaction of a hand, the hand region of SI could reflect the intensity of the observed reaction by simulating the movements of the hand and/or the somatosensory consequences of the harm (*Keysers et al., 2010*). If we know that the painful stimulation originates from the hand, but the intensity of the pain has to be inferred from the facial expressions, somatosensory activation in the hand region could still reflect the intensity of the stimulation, through an indirect route in which information derived from the facial expression is referred back onto the hand region through a process akin to somatosensory imagery, or activation could instead fail to reflect pain intensity. Either way, we could expect the more ventral representation of the face in SI to be involved in representing the intensity of the facial expression (*Adolphs et al., 2000*; *van der Gaag et al., 2007*), but less involved in representing the intensity of a painful hand movement when the face is not visible.

To shed further light on the properties of the hand region of SI in the decision-making task presented in this study, we therefore designed two types of input stimuli that probe the above-mentioned scenarios, both showing different intensity of pain (*Videos 1*, *2*, *3* and *4*) in order to look at quantitative relations between brain activity, perception and behavior (*Wager et al., 2013*). Specifically, participants witnessed a confederate receiving a noxious stimulation of randomly selected intensity (InputMovie) delivered as (a) a swat with a belt on the right hand, with only the hand reaction visible (Hand condition, *Videos 1* and *2*) or (b) an electroshock on the right hand with a visible facial expression and no hand movement (Face condition, *Videos 3* and *4*). In both kinds of videos, the confederate's right hand receiving the painful stimulation was clearly visible on the screen, but in the case of the belt (Hand condition) the reaction of the hand itself was the only cue for the

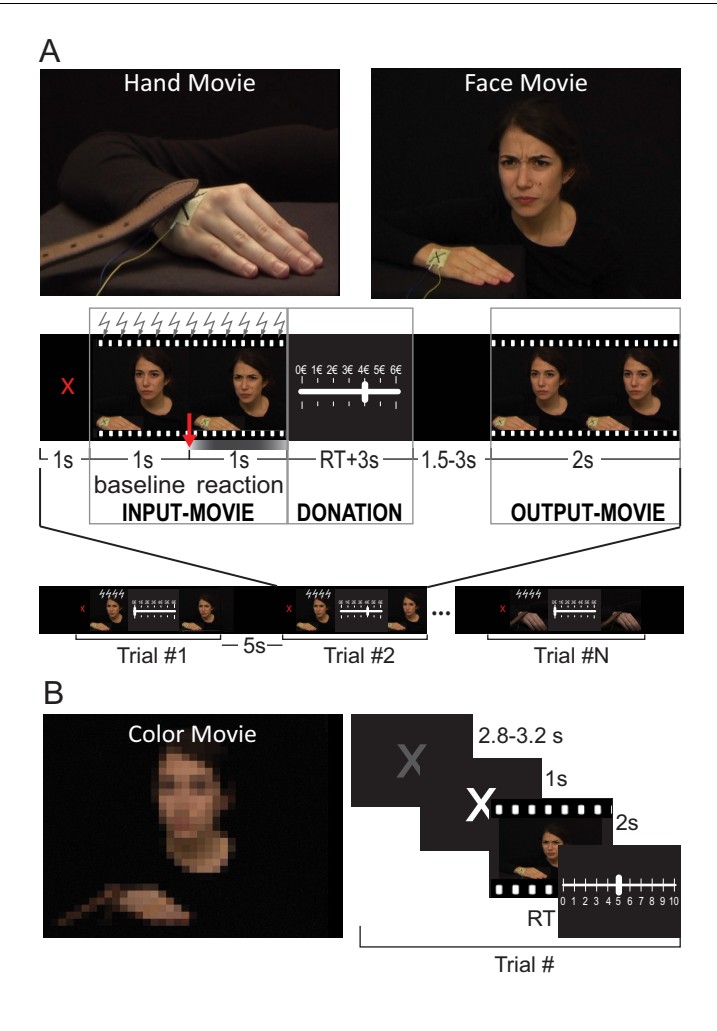

**Figure 1.** Paradigm. (**A**) *Top*: a snapshot from the Hand and Face videos (examples of each condition are presented in ***Videos 1–4***). *Middle*: trial structure. The red arrow indicates the timing of the shock delivery, belt touching the hand or beginning of the color saturation changes. The gray gradient graphically illustrates the dynamic of the face reaction and color saturation changes, with stronger gray corresponding to stronger facial expression or stronger saturation. The intensity of the OutputMovie is equal to the intensity of InputMovie minus the donation. *Bottom*: run structure. The same structure was used in the EEG and TMS experiments. Gray lightning symbols indicate when TMS was applied in the TMS version of the experiment. (**B**) A snapshot from the Color videos (see ***Videos 5***, ***6***) and the trial structure for the rating task.

DOI: https://doi.org/10.7554/eLife.32740.003

The following source data and figure supplements are available for figure 1:

**Figure supplement 1.** Histograms of participants' responses for the credibility and movie repetition detection for the experiment with the Costly Helping paradigm.

DOI: https://doi.org/10.7554/eLife.32740.004

**Figure supplement 1—source data 1.** Participants' responses for the credibility and movie repetition detection for the experiment with the Costly Helping paradigm.

DOI: https://doi.org/10.7554/eLife.32740.005

participants to deduce the painfulness of the stimulation. By contrast, when receiving the electro-shock (Face condition), the hand did not show any reaction and the painfulness was deduced only by the confederate's facial expression. At each trial, participants then received an endowment of 6€ and could reduce the intensity of the next noxious stimulation (OutputMovie) by giving up some of

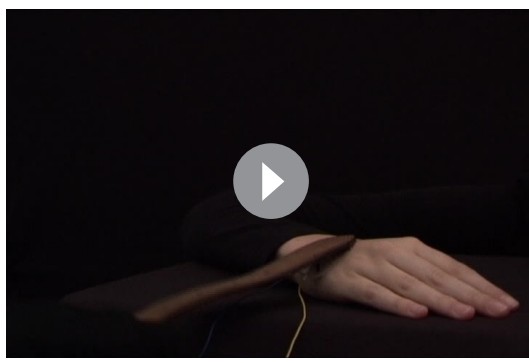

**Video 1.** Example of Hand video of intensity 1.
DOI: https://doi.org/10.7554/eLife.32740.006

that money, knowing that the remainder would be part of their compensation (*Figure 1A* and *Videos 1–6*).

First, we investigated whether activation of the hand region of the left SI, as measured with EEG, explains prosocial behavior. The SI hand region was identified in an independent pool of participants by correlating fMRI BOLD responses within SI with subjective experience of pain elicited by electrical stimulations on the participant's right hand. We hypothesized that activation of the hand region of SI would correlate with the donation given by the participants in the Hand condition, when the intensity of the stimulation had to be deduced from the hand movement. In the Face condition, we predicted that activity more ventrally in SI, where facial expressions are represented, would correlate with the donation. We were able to make this prediction because the relevance of facial mimicry has been highlighted in many studies (*Oberman et al., 2007*; *Hess and Fischer, 2013*; *Fischer and Hess, 2017*; *Wood et al., 2016b*), because we know that the ventral somatosensory cortex causally contributes to emotion perception from facial expression (*Adolphs et al., 2000*; *Paracampo et al., 2017*) and because we know that the emotions from visually presented facial expressions requires ventral somatosensory-related cortices (*Adolphs et al., 2000*). As mentioned above, for the hand region of SI during the Face condition, we had less defined predictions: the presence or absence of correlation of SI hand region activity with the donation while perceiving facial expressions will inform whether facially deduced pain intensity is re-represented in the SI locations reflecting the inferred origin of that pain.

In a second experiment, we then perturbed the SI activity of the hand region with repetitive TMS (rTMS) to test whether disturbing SI vicarious activity altered prosocial behavior. We expected TMS over the hand-region of SI to disrupt the accuracy with which participants can transform the observed kinematics of the belt and hand into an accurate perception of how painful this particular stimulation was for the other. We had this expectation because disrupting SI activity using TMS or neurological lesions has been shown to alter the accuracy with which participants perceive some emotions (*Adolphs et al., 2000*; *Paracampo et al., 2017*) and hand actions (*Valchev et al., 2017*), and because, in the nociceptive literature, SI has been associated more with perceptual than with motivational processes (*Keysers et al., 2010*; *Lee and Tracey, 2010*). We thus expect decision-making to become noisier, and less attuned to the level of pain experienced by the other on a trial-by-trial basis, particularly when the reaction of the hand is the only source of information on which the decision is based (Hand condition). This effect would be weaker or absent when information is derived from the Face, where alternative sources of information are available.

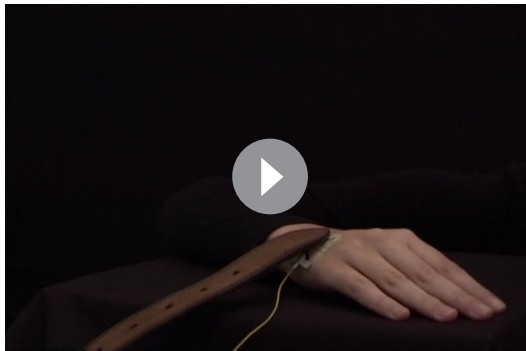

**Video 2.** Example of Hand video of intensity 6.
DOI: https://doi.org/10.7554/eLife.32740.007

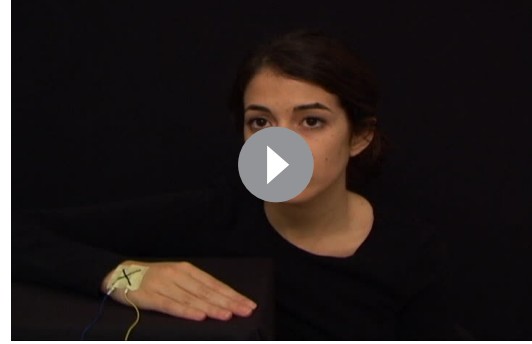

**Video 3.** Example of Face video of intensity 1.
DOI: https://doi.org/10.7554/eLife.32740.008

Finally, we used data from a third experiment to explore whether a disruption of the perception of pain intensity does indeed mediate how disrupting SI activity alters decision-making. Brain activity in SI was altered using high-definition tDCS while participants had to rate how much pain the person in the Hand and Face movies experienced on a trial-by-trial basis. Because the specific montage used in this experiment was expected to facilitate brain activity under the anode placed over the SI hand region and to inhibit brain activity under the return cathodes, one of which was placed over the face region of SI, we expected the accuracy of the ratings to increase in the Hand stimuli and to decrease in the Face stimuli.

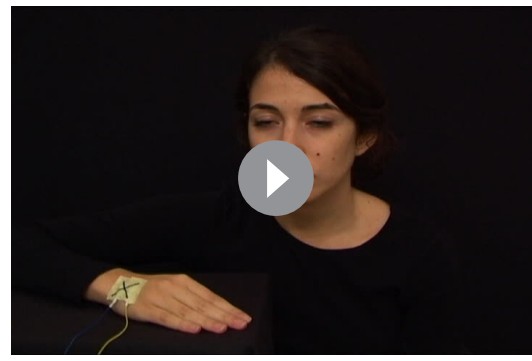

**Video 4.** Example of Face video of intensity 7.
DOI: https://doi.org/10.7554/eLife.32740.009

## Results

### Experiment 1: EEG study

Participants (*Table 1*) donated on average the same amount in the Face and Hand conditions (Face: M = 2.14€, SD = 1.2; Hand: M = 2.16€, SD = 1.2; dependent sample t-test $t_{(28)}$=–0.2, p=0.8), but comparing the standard deviation in donation within each participant showed more variability in donation for the Face condition (Face: $M_{SD}$ = 1.47, $SD_{SD}$ = 0.44; Hand: $M_{SD}$ = 1.22, $SD_{SD}$ = 0.40; dependent sample t-test $t_{(28)}$=4, p=0.0004). To avoid this confound in further analysis, we Z-transformed the donation of each participant separately for the two conditions.

To assess whether participants' donation was driven by the intensity of the reaction shown in the InputMovie, for each participant, we performed a robust linear regression (*Holland and Welsch, 1977*) between the intensity attributed to the movies by an independent pool of participants and the Z-donation. The analysis confirmed that participants' Z-donation closely followed the pain intensity shown in the InputMovies (*Figure 2A*). In the Face condition, all participants had regression slopes that were positive and significantly different from zero (Face slope: M = 0.48, SD = 0.6, average t value for Face slope = 21.9, SD = 17, all p<0.05; group one-sample t-test on Face slopes t (28) = 37.9 p=0.0006E-21). In the Hand condition, one participant had a negative slope but the regression was not significant (t = −0.17, p=0.8), a second participant had a positive but not significant slope (t = 1.89, p=0.06), whereas the remaining participants all showed positive and significant slopes (all p<0.05). We considered the former two as normal variation along the population spectrum and kept them in the analysis (Hand slope: M = 0.45, SD = 1.4; average t value for Hand slope = 7.4, SD = 5.4; group single sample t-test on Hand slopes $t_{(28)}$ = 16.3 p=0.0001E–11). Importantly, the average slope did not differ between the two conditions (paired t-test $t_{(28)}$=−1 p=0.3).

To interrogate the electrical activity originating from the primary somatosensory cortex, we used a linear constrained minimum variance beam-forming approach (*Van Veen et al., 1997*), in which spatial filters were designed to isolate brain electrical activity from the specified locations of interest. To identify regions of SI that reflect perceived pain intensity while participants experience pain on their own bodies, we performed an independent fMRI experiment in which participants received electrical stimulation

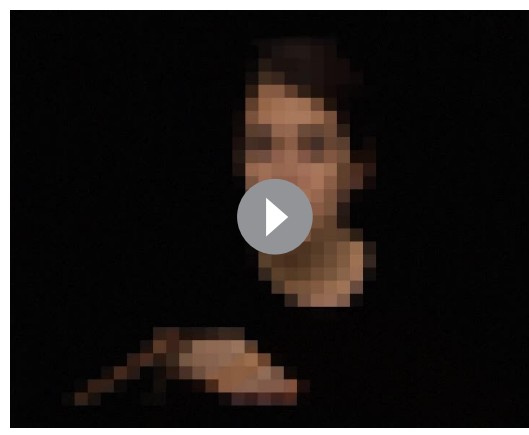

**Video 5.** Example of Color video of intensity 1.
DOI: https://doi.org/10.7554/eLife.32740.010

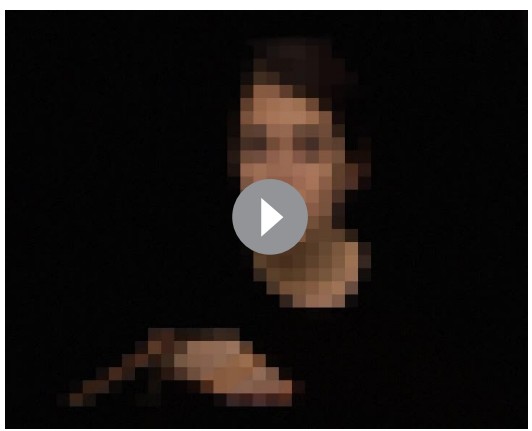

**Video 6.** Example of Color video of intensity 6.
DOI: https://doi.org/10.7554/eLife.32740.011

at different intensities on their right hand and reported how painful each stimulation was. We then identified voxels in the left SI where the BOLD signal correlated with reported painfulness (see 'Supplementary information, Pain localizer experiment'), and used the following clusters as ROIs for the EEG beam-forming analysis: a dorsal (dSI-L; peak at $MNI_{(x,y,z)} = -30,-36, 62$) and a ventral (vSI-L; peak at $MNI_{(x,y,z,)} = -54,-19, 32$) somatosensory cluster (*Figure 2B*; *Maldjian et al., 1999*; *Mancini et al., 2012*). The ventral cluster has a dorso-ventral extent similar to the face representation in SI but seems unusually posterior (*Figure 2B*; *van der Gaag et al., 2007*). This suggests that the ventral cluster could originate from the posterior parietal region PF or could represent the facial reaction to the pain in SI. We focused on the left hemisphere, because electrical stimulation was always delivered to the right hand of both the confederate shown in the movies, and of the participants in the pain localizer study.

For each participant and ROI, beam-forming returned activity time-courses along three standard dipole directions. Given the low spatial resolution of EEG, the mixed orientation of cells in the somatosensory cortex (encompassing gyri and sulci), and the fact that we use videos, for which the dipole capturing most of the variance could change over time, we included all three dipole directions in multivariate analyses.

To assess whether EEG activity explained variance in participants' donations, we used a random effect, summary statistics approach routinely used in fMRI analysis (*Holmes et al., 1998*). First, at the single subject level, we modelled the relationship between SI ROI activity and donation by calculating a robust regression between brain activity at a given time-point and the Z-donation for all the trials of that participant (*Figure 3A*, left). Repeating this analysis for each time point generated a time-course of the slope separately for each participant, condition and dipole (*Figure 3A*, right). Second, at the group level, we analyzed the group distribution of these slopes: if EEG activity in an ROI does not carry systematic information about the donation, the slopes would be randomly distributed around zero. To test this null hypothesis, we used the Hotelling's t-squared statistic, a

**Table 1.** Participants characteristics.
The table indicates the number of tested participants for each experiment, with those excluded from the analyses within brackets; the average age and its standard deviation (SD); the gender ratio; and the experimental task. Three participants from the EEG and three from the TMS experiment were excluded because they did not sufficiently believe the cover story. One participant in the EEG was excluded because of EEG failure. One participant in the tDCS was excluded because they performed at chance level. Analyses on gender effects can be found in *Supplementary file 3*.

| | Total N° subj. (excluded) | Age (SD) | Gender M /F | Experimental task |
|---|---|---|---|---|
| Validation costly helping stimuli | 40 | 24 (6) | 23/17 | Rating other's pain |
| Validation rating stimuli | 20 | 24-(3.4) | 8/12 | Rating other's pain and color saturation |
| fMRI | 25 | 25 (6) | 11/14 | Rating own pain |
| EEG | 32 (4) | 25 (5) | 16/16 | Decision to help |
| TMS | 18 (3) | 25 (7) | 12/6 | Decision to help |
| HD-tDCS | 26 (1) | 25 (4) | 13/13 | Rating other's pain and color saturation |

DOI: https://doi.org/10.7554/eLife.32740.012

The following source data available for Table 1:

**Source data 1.** Participant's demographic information.
DOI: https://doi.org/10.7554/eLife.32740.013

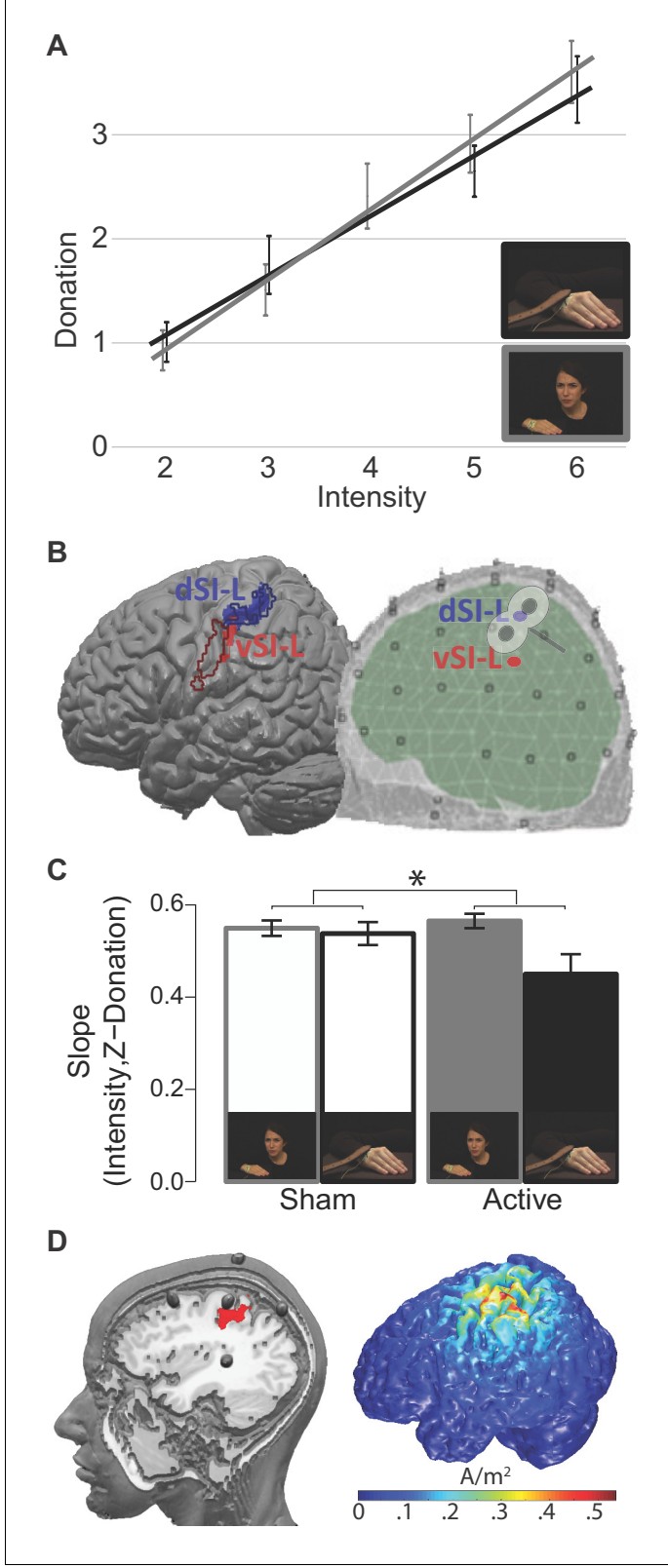

**Figure 2.** ROIs and TMS results. (**A**) The relationship between InputMovie intensity, as assigned by an independent pool of participants during the movies validation procedure, and given donation for hand and face videos. Each point is the group average donation for the specific intensity. Error bars represent S.E.M. (**B**) Pain Localizer ROIs. *Left:* results of the pain localizer within the primary somatosensory cortices

*Figure 2 continued on next page*

*Figure 2 continued*

(see 'Supplementary information') shown on the Colin brain together with contours of regions associated with hand (blue) and face (red) movements. These contours were generated using the meta-analyses tool Neurosynth (*Yarkoni et al., 2011*). Specifically, we generated reverse inference maps using the search terms 'grasping' and 'speech production' to probe movements of the hand and of the face, respectively, and intersected each with an anatomical map of the left SI from the anatomy toolbox (as the union of BA1, 2, 3a and 3b). *Right:* schematic visualization of the dorsal and ventral ROIs within the EEG template space, and approximate site of the TMS stimulation. (C) Interaction Condition x TMS results. *p<0.05. Error bars represent S.E.M. (D) The left render shows the location of the five HD-tDCS electrodes on the scalp and where the central anode is positioned relative to our d-SIL ROI (red). The image was created by inserting fish oil omega three pills in place of the HD-tDCS electrodes inside the electrodes holders. A participant was wearing the montage while a T1-weighted anatomical image was acquired (TR = 8.2 ms, TE = 3.8 ms, flip angle = 8°, FOV = 240 mm × 240 mm, 1 × 1 × 1 mm isotropic voxels). The right render shows the 3D simulation of current density changes expected from our tDCS montage, obtained using the electrostatic finite element method (FEM) offered by the Matlab toolbox COMETS 2 (*Jung et al., 2013*).
DOI: https://doi.org/10.7554/eLife.32740.015

The following source data is available for figure 2:

**Source data 1.** Average donation given by participants for the specific intensity and condition.
DOI: https://doi.org/10.7554/eLife.32740.016

**Source data 2.** dSI-L and vSI-L ROIs and BEM model used in *Figure 2A*.
DOI: https://doi.org/10.7554/eLife.32740.017

**Source data 3.** Single participants' robust slopes between the intensity of Input Movie and given donation in the TMS study.
DOI: https://doi.org/10.7554/eLife.32740.018

multivariate generalization of the Student's t-test that combines evidence from all three dipoles and all participants in a given condition and time-point. We controlled for multiple time point testing using a cluster-based randomization test implemented in Field-Trip (*Oostenveld et al., 2011*), which compares the sum of the Hotelling statistics within the clusters in the real data against those in clusters obtained after switching the sign of the entire slope-time course of randomly selected participants. We repeated the procedure for each condition and ROI and accounted for those additional comparisons using a Bonferroni procedure. Only results surviving those corrections are reported in yellow in *Figure 3B*. Results show that variation in activity in dSI-L explained variation of Z-donation in the Hand condition in the time windows between 420 and 452 ms (cluster-statistic = 142.2, p=0.002) and between 458 and 476 ms (cluster-statistic = 91.4 ms, p=0.006) after the belt hits the confederate's hand. Activity in dSI-L also related to Z-donation in Face condition but later in time, between 516 and 602 ms (cluster-statistic = 602.5 ms, p=0.0009), between 608 and 754 ms (cluster-statistic = 333.8 ms, p=0.0009), and also between 808 and 880 ms (cluster-statistic = 330 ms, p=0.0009) after the noxious stimulation was delivered. vSI-L brain activity explained Z-donation but only in the Face condition in the time window between 402 and 430 ms (cluster-statistic = 102 ms, p=0.005; *Figure 3B*).

To visualize this effect, we categorized the trials into low and high donation categories (median split per participant) and calculated grand-averages voltage time courses during the InputMovie (*Figure 3C*) for each dipole. The grand averages along Y and Z present a negative deflection after the InputMovie onset independently of condition and donation, which probably reflects general attentional processes. For the Hand condition, the dipoles further present a positive peak some milliseconds after the slap, which is sustained along Z and transient along X and Y. No clear peak is recognizable after the shock (time 0) during the Face condition. Differences between low and high donation responses can be observed, in particular along Y, for all ROIs and conditions (*Supplementary file 1* and *2*).

There is evidence for a bilateral receptive field in the Brodmann 1 and 2 sub-regions of SI (*Iwamura et al., 2002*), and for the involvement of the right hemisphere in the perception of emotion (not including pain) from facial expressions (*Adolphs et al., 2000*); bilateral activation reported by *Ashar et al., 2017*; *Lamm et al., 2011*; *Cui et al., 2015*; *Carr et al., 2003*), and from hand movements (*Christov-Moore and Iacoboni, 2016*). *Figure 3D* shows the signal originating from mirroring our left ROIs onto the right hemisphere, and in yellow, the time points significantly explaining the

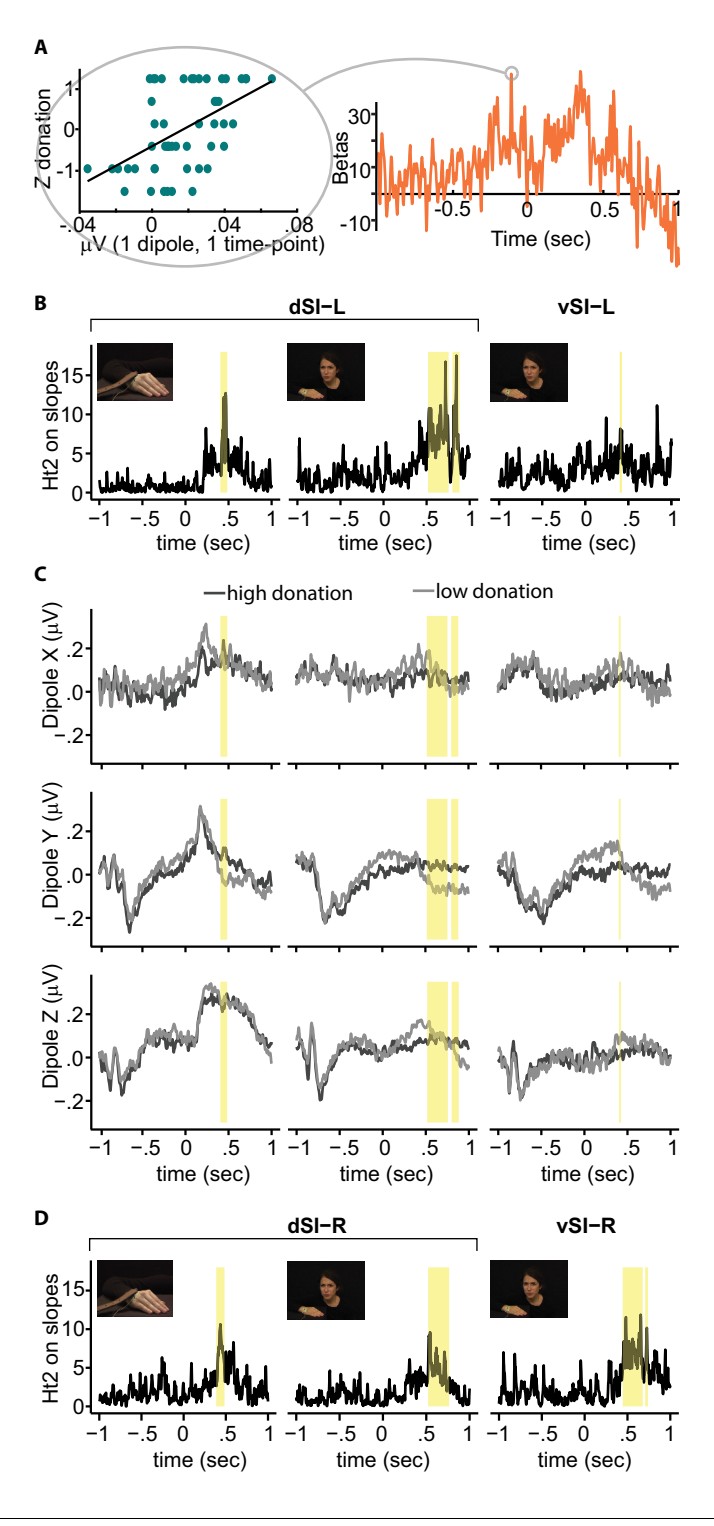

**Figure 3.** Regression between SI activity and donation. (**A**) *Left:* relationship between brain activity of one example participant at a given time-point and the Z-donation for all the trials of that participant. The linear trend represents the slope of the robust regression performed on these values. *Right:* time-course of the robust regression slopes (betas) for the same example participant. (**B**) Time-course of the Hotelling's t-squared (Ht2) test on the slopes for the significant ROI and condition. Because the two significant Hand clusters are very close in time, for illustrative purposes only, they have been evidenced by a single yellow band. (**C**) Grand averages for high (darker lines) and

*Figure 3 continued on next page*

*Figure 3 continued*

low (lighter lines) donation for each dipole, SI-ROI and condition. (**D**) Right hemisphere results. Significant clusters based on Ht2 are shown in yellow.

DOI: https://doi.org/10.7554/eLife.32740.019

The following source data is available for figure 3:

**Source data 1.** Brain activity of one example participant at a given time-point and the Z-donation for all the trials of that participant.
DOI: https://doi.org/10.7554/eLife.32740.020
**Source data 2.** Time-course of the Hotelling's t-squared (Ht2) test on the slopes for the significant ROI and condition.
DOI: https://doi.org/10.7554/eLife.32740.021
**Source data 3.** Group average of the brain activity in each time point and condition.
DOI: https://doi.org/10.7554/eLife.32740.022
**Source data 4.** Time-course of the Hotelling's t-squared (Ht2) test on the slopes for the mirror ROI and condition.
DOI: https://doi.org/10.7554/eLife.32740.023

donation. For the hand region of SI (d-SI), results for the two hemispheres are very similar, suggesting a lack of clear hemispheric specificity (*Figure 3D* and 'Supplementary Information'). For the more ventral, putative face region of SI (v-SI), responses appear stronger on the right hemisphere, in line with previous findings (*Adolphs et al., 2000*).

Our EEG findings therefore suggest that while witnessing the pain of another person, the magnitude of brain activity in the hand region of SI (d-SI) could inform decision-making. To examine the causal contribution of this region to decision-making, in a second experiment, we used TMS to disturb the activity of the SI hand region. We targeted the left hemisphere because it is contralateral to the hand that is stimulated in the confederate.

## Experiment 2: TMS study

A separate group of participants (*Table 1*) performed the costly helping paradigm for the Face and Hand conditions under both active and sham rTMS over the left SI hand region.

The within-subject ANOVA with two factors, Condition (Face and Hand) and TMS (Active and Sham), on the standard deviation calculated across the donation of each participant reveals that participants varied their donations from trial-to-trial more in the Face than in the Hand condition (main effect of condition $F_{(1,14)} = 98$, p=0.001E–4), but there was no variance difference between the two TMS protocols (main effect of TMS $F_{(1,14)} = 0.2$, p=0.65, interaction between Condition and TMS $F_{(1,14)} = 0.2$, p=0.7). Therefore, the same Z-transformation of the donation was used as in Experiment 1, standardizing separately all of the donations of the Hand and Face conditions of each participant (but without separating Active and Sham to preserve TMS effects).

TMS had no effect on the average Z-donation in either task, as indicated by the repeated ANOVA with factor Condition (Face and Hand) and TMS (Active and Sham): TMS $F_{(1,14)} = 2.3$, p=0.15, TMS x Condition $F_{(1,14)} = 0.5$, p=0.56. To test whether TMS interferes with the relationship between the intensity of the movie and the donation, for each participant we calculated the slope between the intensity of the movie and the Z-donation by means of a robust regression, separately for the Condition (Face and Hand) and the TMS protocol (Sham and Active).

We then performed a two-factor repeated ANOVA on these slopes, using the above-mentioned two-level summary statistic approach. The analysis showed that the slopes for the Face condition are steeper than those for the Hand (Face: M = 0.56, SD = 0.06; Hand: M = 0.49, SD = 0.14; Main effect of Condition: $F_{(1,14)} = 7$, p=0.02). TMS did not have a general effect common to the two conditions (Main effect of TMS: $F_{(1,14)} = 1.9$, p=0.19). Interestingly, the ANOVA showed a significant interaction effect ($F_{(1,14)} = 4.8$, p=0.04, partial $\eta^2 = 0.25$; *Figure 2C*) with a larger TMS effect in the Hand compared to that in the Face condition. A Newman-Keuls post-hoc test indicated that active rTMS on SI significantly flattened the relationship between intensity and donation only for the Hand condition (Hand Sham M = 0.54, DS = 0.1; Hand TMS M = 0.45, DS = 0.2; p=0.02). There is no evidence for such an effect in the Face condition (Face Sham M = 0.55, SD = 0.06; Face TMS M = 0.56, SD = 0.06; p=0.6). To test whether the lack of effect in the Face condition was due to limited statistical power or whether it provides evidence for the null hypothesis ($H_0$) of no (sizable) effect of TMS,

we used Bayesian statistics. We calculated an index of TMS effect for each participant in the Face condition as follows: slope in the Active session – slope in the Sham session. We then performed a Bayesian one sample t-test using JASP with default priors (https://jasp-stats.org) that showed that the null hypothesis is 5.7 times more likely than the alternative $H_1$ (Bayes factor $p[H_0$:index $\geq 0|$ data]/$p[H_1$:index $<0|$data]=5.7), providing positive evidence for the absence of a sizable effect in the Face condition (*Kass and Raftery, 1995*).

The effect on the Hand condition did not correlate with any of the TMS side effects perceived by the participant while performing the experiment (measured by questionnaire, all $p>0.05$). This suggests that TMS on the SI hand representation interferes with the process that normally couples a person's donation to the needs of the other person (i.e. the observed pain intensity) when this need is perceived through the movements of the affected body part (Hand condition), but this is less the case when the need is perceived through facial expressions. To test whether the impact of TMS is mediated by an effect on pain perception, we used data from a third experiment.

## Experiment 3: HD-tDCS study

In this experiment, participants (*Table 1*) had to rate how much pain they perceived while watching the Face and Hand videos under the effect of tDCS centered over left SI. The high density 4 × 1 electrodes tDCS montage that we used would be expected to have a facilitatory effect on the hand region of SI under the anode (*Figure 2D*), and a weaker inhibitory effect on ventral SI, including the face representation, under one of the four return cathodes. To control for unspecific effects on intensity rating processes unrelated to pain in this experiment, we introduced a new type of video in which participants needed to rate color saturation intensity (Color condition, *Figure 1B*).

A 2 Stimulation (tDCS and Sham) x 3 Condition (Face, Hand and Color) repeated measures ANOVA on the standard deviation calculated for the participants' rating reveals that participants did not use the rating scale differently in the three conditions (main effect of condition $F_{(1,24)} = 0.4$, $p=0.5$), nor between the two tDCS sessions (main effect of Stimulation $F_{(1,48)} = 2$, $p=0.1$, interaction between Condition and Stimulation $F_{(1,48)} = 0.6$, $p=0.5$). To be consistent between our studies, we applied Z-transformation of the ratings separately for the three conditions but while pooling the two tDCS conditions. A Stimulation (tDCS and Sham) x Condition (Face, Hand and Color) ANOVA on the average rating revealed that average rating remained stable across conditions (Stimulation $F_{(1,24)} = 0.5$, $p=0.5$; Condition $F_{(1,48)} = 0.8$, $p=0.5$; Interaction $F_{(1,48)} = 0.3$, $p=0.7$). For each condition and session, we then correlated the intensity assigned to the movies during the validation process and the Z-rating given by the participant. The correlation values were normalized using the Fisher z-transformation. We then performed a two-factor repeated ANOVA on the correlation coefficients obtained. The analysis showed that the correlation coefficients differed between conditions (Face: M = 1.3, SD = 0.2; Hand: M = 1, SD = 0.2; Color: M = 1.2, SD = 0.3; Main effect of Condition: $F_{(2,48)} = 17$, $p<0.002E–3$). A post-Hoc Newman-Keuls test revealed that the Hand condition had a lower correlation coefficient than both the Face and Color conditions, while the latter did not differ from each other. tDCS had no main effect on the correlation coefficients (main effect of tDCS $F_{(2,48)} = 0.5$, $p=0.5$). Interestingly, tDCS had a different effect depending on the conditions (Interaction: $F_{(2,48)} = 3.4$, $p=0.04$, $\eta^2=0.12$). Planned paired t-test comparison between sham and tDCS session for each condition indicated that the tDCS on SI significantly improved the relationship between intensity and rating for the Hand condition (Hand Sham: M = 1, DS = 0.2; Hand tDCS: M = 1.1, DS = 0.2, $t_{(24)} = –2$, $p=0.04$), while it showed a trend for reduction in the Face condition (Face Sham: M = 1.3, DS = 0.2; Face tDCS: M = 1.2, DS = 0.2; $t_{(24)} = 1.8$ $p=0.07$), and no appreciable change in the Color condition (Color Sham: M = 1.3, DS = 0.2; Color tDCS: M = 1.2, DS = 0.3; $t_{(24)} = 1$, $p=0.3$). Again, to test whether the lack of effect in the Color condition supports the null hypothesis, we calculated an index of the stimulation effect by subtracting the z-transformed correlation score calculated in sham from the one calculated in the real tDCS session. As Hand and Face conditions showed opposite effects, we performed a 2-tailed one sample Bayesian t-test on the Color condition. The Bayes factor indicated that the $H_0$ was 3.1 times more likely than the $H_1$, confirming that the Color condition does not change after HD-tDCS. The Hand and Face effects did not correlate with any of the tDCS side effects perceived by participants while performing the experiment (all $p>0.05$).

# Discussion

We used a helping task in which brain activity could be related to prosocial behavior on a trial-by-trial basis, and concentrated on measuring and altering activity in the hand region of SI, to shed light on the contribution of this region to prosocial decision-making.

Specifically, we localized regions in the left SI that encode the intensity of pain experienced by a group of participants using fMRI. This evidenced two clusters in the left SI: a dorsal cluster corresponding to the hand representation of SI and a ventral cluster that had a dorso-ventral extent similar to the face representation of SI. These ROIs served as the ROIs for our EEG experiment, which showed that the magnitude of brain activity originating from the dorsal cluster had a significant relationship with helping. This was true whether victims expressed their reaction through their afflicted body part or through their face. *Figure 3* shows how the timing of that activity followed the timing of the information in the movies: the hand immediately retracts at the moment the belt hits the hand, and SI activity had a sharp and sudden peak in explanatory power, whereas the facial expression develops more slowly after the shock is delivered, and SI activity showed a more progressive and sustained explanatory power. *Figure 3* also shows that the ventral and dorsal sector of our functionally localized SI nociceptive representation behave differently in our task, with the dorsal (hand) ROI explaining donation for either source of information (Face or Hand) and the ventral (face) ROI being explanatory only for facial expressions.

The choice to interrogate the signal during pain observation within regions coding the intensity of pain during self-pain experience was, as mentioned in the introduction, dictated by the theoretical framework of emotional contagion and vicarious activity. In this framework, the activation of cells involved in experiencing pain during the observation of other people's pain would help someone to 'feel' what another person is experiencing by inducing a psychological state similar to that to which these neurons contribute during the experience of pain. FMRI overlaps between the experience and observation of pain have been widely documented, and taken as support for such a framework (*Keysers et al., 2010*; *Lamm et al., 2011*). This notion was recently in the focus of debate because of mixed results from multi-voxels pattern analyses (*Zaki et al., 2016*; *Corradi-Dell''Acqua et al., 2016*; *Krishnan et al., 2016*). The logic of these analyses is to identify a pattern across voxels that discriminates different intensities of experienced pain from fMRI signals. If pain observation triggers a neuronal representation of felt pain, the logic goes, the same pattern should discriminate different intensities of observed pain — so-called above-chance cross-modal classification. Some scientists find this to be true (*Corradi-Dell''Acqua et al., 2016*), others not (*Krishnan et al., 2016*). It is important to realize, however, that a region may have neurons involved in experiencing and observing pain, as our framework predicts, without significant cross-modal classification of fMRI signals (*Zaki et al., 2016*): decades of work on mirror neurons for actions show that only 10% of neurons involved in performing an action become recruited while observing that action (*Gallese et al., 1996*; *Keysers et al., 2003*; *Mukamel et al., 2010*). If the same is true for pain, only 10% of neurons involved in pain experience may also be recruited during pain observation. This low percentage means that a pattern classifier trained on pain experience would be dominated by the signals originating from the 90% of neurons that are not involved in pain observation, and would then fail to interrogate the 10% involved in observation reliably when tested with pain-observation data.

Until we have systematic single-cell data during the experience and observation of pain (*Hutchison et al., 1999*), it will not become clear whether neurons represent felt and observed pain reliably (*Zaki et al., 2016*). Accordingly, our finding that signals from a region of SI involved in actual pain experience explains variance in helping in our paradigm is compatible with the notion that this signal originates from neurons also involved in pain experience. Alternatively, the signal could originate from neurons not involved in pain experience that are simply spatially intertwined with those involved in pain experience (*Zaki et al., 2016*; *Keysers and Gazzola, 2009*). In this manuscript, when we speak of vicarious pain activations, we therefore mean activations of regions involved in pain experience during the observation of the pain of others, without having the means to assess whether this activation originates from the neurons that are involved in actual pain experience.

When pain is expressed by the reaction of the hand, the dorsal SI hand region activation correlates with decision-making (EEG). In addition, altering this activity changes (TMS) decision-making, suggesting that the SI hand region activation feeds into the decision-making process. When pain is expressed by the face, the SI hand region activation correlates with decision-making (EEG) but

altering this activity influences decision-making less. This latter finding is compatible with three interpretations. (i) Activity in the hand-region of SI is an epiphenomenon, representing an imagination of what the painful stimulation would have felt on the hand (*Fairhurst et al., 2012*), which is not used in decision-making. (ii) The SI hand region activity is used for decision-making in the sham condition, but can be substituted by alternative sources of information derived from the face elsewhere in the brain in the active TMS condition. Both of these interpretations are reminiscent of the notion that pain information takes different paths depending on the stimulus it is derived from (*Keysers et al., 2010*; *Lamm et al., 2011*). (iii) Information in the SI hand region has a higher signal to noise ratio in the Face condition than in the Hand condition, making it less susceptible to TMS interference.

Our results show that trial-by-trial amplitude of the EEG activity from ventral SI significantly explains changes in donation in the Face condition only. This effect could be driven by a covert internal simulation of the other's facial expression or by overt facial mimicry. Interfering with facial mimicry has been shown to impair visual recognition of expressions (*Oberman et al., 2007*; *Wood et al., 2016a*) and interfering with activity in ventral somatosensory cortex alters the recognition of emotion from faces (*Adolphs et al., 2000*; *Paracampo et al., 2017*). Future research should neuro-modulate brain activity in ventral SI in addition to the hand representation we targeted here, while measuring the willingness to help in order to further investigate the dissociation suggested by our results. The emergence of focused ultrasounds as a focal neuro-modulation method (*Mueller et al., 2014*; *Lee et al., 2015, 2016*) could enable such studies without the muscle artefacts that are inevitable with TMS. HD-tDCS, as used in our third experiment, also has the advantage of not causing muscle twitches, but lacks the focality needed to allow a confident argument that one can disentangle the contributions of the face and hand regions that are located only 2 cm apart.

In the Hand movie, movements are displayed both in the first half of the video by the agent wielding the belt and in the second half by the victim's hand being compressed by and reacting to the swat. Activity in SI has been shown to have the potential to encode all of these (*Keysers et al., 2010*). Interestingly, SI activity significantly predicted the donation only during the second half, in which the victim's hand is compressed by the belt and reacts to it. This suggests that it is SI's ability to represent the impact of the belt on the hand or the reaction of the victim to the stimulation on the hand that induced the prosocial decision-making.

Furthermore, we addressed the issue of how SI contributes to decision-making, leveraging a third HD-tDCS experiment that allowed us to discriminate between perceptual or motivational contributions. Our results suggest that SI activation in the hand region contributes to prosocial decision-making by transforming the sight of hand-movements caused by a swat into a perception of pain-intensity, which then serves as an input to a decision-making process elsewhere. If this trial-by-trial perception is perturbed, our decision to help no longer optimally follows the trial-by-trial variance in pain experienced by others. This function is similar to the function that SI has during the observation of actions. For instance, disturbing the activity of the SI hand region with TMS makes ratings of the weight of an object seen lifted noisier than in a sham condition, suggesting that the region is necessary for transforming observed hand kinematics into an estimate of the forces that have been acting on the hand (*Valchev et al., 2017*). Similar kinematic analysis may underpin the transformation of the observed hand kinematics following the swat into a painfulness estimate in our Hand condition. Affective social reactions, be they personal distress or empathic concern, would be informed by this kinematic analysis in SI, but require additional processes that the pain experience literature would ascribe to the anterior insula and cingulate (*Lee and Tracey, 2010*). In this interpretation, a neural network including SI informs the participant on how intense the swat was on a given trial, and determines the ability of the participant to adjust the donation to the circumstances of a specific trial. By contrast, the mean donation could reflect trait differences in empathic concern (measured by the Interpersonal Reactive Index [*Davis, 1983*]) and money attitude (*Yamauchi and Templer, 1982*) ('Supplemental analysis — Correlation with self-reported questionnaires'). A more in-depth understanding of what emotional feelings (pain-like personal distress vs. more positively valenced empathic concern) accompany the motivational effect of SI activation on high-pain trials remains unclear from our data, and could be studied in future research by asking participants to provide specific ratings of their own affect on a trial-by-trial basis.

In summary, we provide evidence that activity in the hand region of SI that occurs while witnessing the bodily reactions of a victim to a painful stimulation is not only correlated with the willingness to help but significantly influences prosocial decision-making. Our data further constrain the

mechanisms through which SI influences prosocial decision-making by showing that altering SI activity also influences the perception of other people's pain intensity. This suggests that the role of SI is to help us transform the kinematics of affected body-parts into a perception of pain, which is then a significant input to a decision-making process that occurs elsewhere in the brain. If pain is not expressed through the affected body part but communicated through facial expressions, SI activity in the somatotopic representation of the initially affected body-part no longer seems to be a necessary input to this decision-making. These neuromodulation findings support the notion derived from neuroimaging literature that multiple networks, depending on the nature of the stimulus, can be recruited during the perception of the pain of others (*Keysers et al., 2010*; *Lamm et al., 2011*). Future studies will be needed to isolate and characterize the causal contribution and interaction across the nodes of these networks, and to further characterize the conditions under which each network is necessary. An empirical foundation for the intuitively attractive and often suggested causal links between the ability to represent what other people feel and prosocial actions is provided by the evidence that SI vicarious activations directly influence prosociality.

## Methods and material

### Participants

A total of 169 healthy, right-handed volunteers, with normal or corrected-to-normal vision, (mean age = 25 +/– 5 SD) were recruited for our studies (*Table 1*). Because previous studies reported that racial biases modulate empathy (*Xu et al., 2009*; *Avenanti et al., 2010*; *Cikara et al., 2014*) and our videos showed a Caucasian confederate, only Caucasian individuals were recruited. All participants received monetary compensation and gave their informed consent for participation in the study. None of the participants reported neurological, psychiatric, or other medical problems or any contraindication to fMRI, TMS or tDCS (*Rossini et al., 2015*; *Rossi et al., 2009*). No discomfort or adverse TMS effects were reported by the participants or noticed by the experimenter. *Table 1* summarizes the number and characteristics of the participants for each study.

All studies have been approved by the Ethics Committee of the University of Amsterdam, The Netherlands (project identifiers: 2016-BC-7394, 2016-BC-7130, 2017-EXT-8467, 2016-PSY-6485, 2014-EXT-3476, and 2014-EXT-3432). All participants received monetary compensation and gave their informed consent for participation in the study. Consent authorization for the publication of images has been obtained.

### Costly helping experimental set-up

Central to our task was the aim of inducing an effective, naturalistic moral dilemma, in which the state induced by witnessing the distress of another individual is pitched against financial rewards. The other person's distress was elicited by delivering electroshocks or slaps on the right-hand dorsum. To limit the total number of shocks or slaps delivered throughout the experiments and to avoid uncontrollable variance in the reactions of the victim, we developed a cover story. Each participant was made to believe that she/he would be paired to another participant, with whom she/he will draw lots to decide who would play the role of the observer and who of the pain-taker. The observer and the pain-taker will be allocated to separate adjacent rooms, connected through a video camera. While the pain-taker would receive the electroshocks and slaps, the observer would have an EEG recorded while they witnessed the reaction of the pain-taker to the stimulations (Experiment 1) or while brain stimulation was delivered over SI (Experiment 2). In reality, the lots were manipulated in such a way that the confederate would always be selected as the pain-taker and the participant as the observer. In addition, participants were misled to think that the noxious stimulations were delivered to the confederate in real-time, and that what the participants saw on the monitor was a live feed from the pain-taker's room. In reality, we presented prerecorded videos of face and hand reactions to noxious stimuli previously delivered to the confederate (*Videos 1*, *2*, *3* and *4*). The confederate's appearance during the experiment was carefully matched to the pre-recorded videos. The exact setup shown in the videos was recreated at every session, including the belt and electric stimulator used, and shown to the participants. Face and Hand videos were shown in separate sessions (order randomized across participants), with a long break in between. During the break, participants

could move, leave the experimental room and, importantly, briefly interact with the confederate. This short break helped to maintain the cover story.

The choice to use a cover story was dictated by (i) pilot data showing that relaxing the cover-story, for instance by acknowledging that the confederate was in fact an experimenter, led to a notable reduction in donation, and (ii) the effort to keep the variance introduced by having different victims at a minimum, facilitating group analyses. In addition, our initial piloting also revealed that testing participants over multiple days led to increased skepticism, which was why we decided to use different pools of participants for each experiment, and why we had to limit the number of trials in the TMS study to what could fit a within subject design.

At the end of the costly helping paradigm, participants answered the question 'Do you think the experimental setup was realistic enough to believe it' on a scale from 1 (strongly disagree) to 7 (strongly agree). Five was used as cut off to discriminate participants who believed in the cover story from those who did not, and participant's who reported four or less were excluded from the analyses. The credibility values for the whole sample of participants are shown in *Figure 1—figure Supplement 1*.

## Costly helping visual stimuli

Two types of 2 s long videos were generated (*Figure 1A*). The Hand-videos depicted the confederate's right hand reactions to a slap delivered by a brown leather belt (procedure adapted from *Meffert et al. [2013]*). The hand, right arm and shoulder are the confederate's only visible body parts. While the belt was visible, the hand holding it only entered the field of view marginally at times, and was covered by a black glove to blend in with the black background. The videos started with the belt laying on the hand dorsum. The slap occurred at the end of the first second, during which the belt would be lifted and prepared to hit. Videos ended one second after the slap, after showing the hand and shoulder reaction. A total of 200 Hand-videos were recorded by varying the intensity of the slap at every trial, with 30 s between each trial. The Face-videos showed the actor's facial expressions in response to an electroshock delivered to the right-hand dorsum. The upper part of her body was clearly visible on a black background. Even though the stimulation was given to her hand and the hand was visible, the hand did not move in response to the shock making the face the main source of information about the stimulation intensity. The videos started with the face in a neutral expression. During the first second, the expression was kept neutral until the stimulation occurred.

Both the hand and face movies were centered not on the moment in which the noxious stimulation was delivered but on the reaction of the actor to them. In the Hand videos, the central frame was the one in which the belt hit the hand with consequent immediate reaction of the hand. In the Face videos, the central frame was the one in which the facial expression began to change (i.e. at +1 s from the beginning of the movie).

A total of 392 Face videos were recorded. The electrical stimulation was a 100 Hz train of electrical pulses of 2 ms pulse duration (square pulse waveform) delivered via a bipolar concentric surface electrode (stimulation areas: 16 mm$^2$). The electrodes were attached to the skin with tape, which was also left in place during the Hand-video recording. A black 'X' was drawn on the tape to clearly show the electrode's position. Each stimulation lasted 1000 ms and varied in current intensity, which ranged from 0.2 mA to 8.0 mA. Thirty seconds were left between stimulations. Current intensity was determined prior to video recording, by following a procedure well established in the literature (*de Vignemont and Singer, 2006*; *Cui et al., 2015*): starting from 0.1 mA, the current was gradually increased until reaching a maximum of 8.0 mA in increments of 0.2 mA. The actor was instructed to evaluate how painful each stimulation was on a 10-point scale, and the current intensities to be used during video recording were chosen according to a maximum perceived intensity of 8. The actor was the same Caucasian woman for all of the videos and experiments: author SG who played the confederate role. The videos were recorded using a Sony DSR-PDX10P Camcorder (Sony, Minato, Tokyo, Japan), and were edited using Adobe Premiere Pro CS6 (Adobe, San Jose, CA, USA).

The 392 videos that were generated were validated by an independent group of 40 participants who did not participate in the other experiments (*Table 1*). Hand and Face videos were presented (*OkazoLab Ltd, 2012*) in separate blocks, counterbalanced across participants. Participants were instructed to observe them and to report the perceived pain intensity that the person in the video felt, using the same 10-point scale employed in the pain-threshold assessment. Average ratings

were computed and rounded off. The results yielded six different movie categories with average pain intensities perceived as 2, 3, 4, 5, 6, and 7 out of 10. Of those, we selected different subsamples for each experiment on the basis of the following two criteria: (a) a low standard deviation in rating across participants ensuring that a specific pain intensity was communicated reliably, and (b) maximized the statistical power of the regression (i.e. privileging movies at the extremes of the intensity range). For the EEG task, we privileged the criterion (b) and included a larger number of trials to reduce neural habituation (95 trials in total). For the TMS experiment, we privileged criterion (a) to maximize the sensitivity to small changes in perceptual accuracy, and only used those movies in which ratings were most concordant (*Table 2*). In both the EEG and TMS experiments, the Face and Hand conditions did not differ in average perceived intensity (EEG: $M_{Face}$ = 3.7, SD = 1.7; $M_{Hand}$ = 4, SD = 1.4; t(93) = 0.88, p=0.4. TMS: $M_{Face}$ = 3.8, SD = 1.6; $M_{Hand}$ = 3.9, SD = 1.2; t-test $t_{(118)}$ = −0.26, p=0.9) or standard deviation (EEG: $F_{(42,51)}$ = 1.49, p=0.09; TMS: $F_{(14,14)}$ = 1.67, p=1.17).

Because the intensity of the OutputMovie depended on the participant's donation, it was impossible to precisely predict the number of videos needed for each intensity and participant. This means that in some cases, the number of recorded videos was lower than the number of actual presentations of a particular intensity, and few videos had to be shown more than once. Care was taken to maximize the distance between repetitions of the same stimulus. The number of repeated videos is low and it was never the case that a participant saw all of the movies twice. During debriefing, we asked our participants to indicate on a seven step scale (from 1 = strongly disagree to 7 = strongly agree) how well the following statement applied to them: 'You think you saw the same movie twice.' On average, participants in the EEG experiment reported a value of 6.1 (*Figure 1—figure Supplement 1*), whereas those in the TMS experiment reported a value of of 6.3, suggesting that movie repetition was not easily recognized by our participants.

## Costly helping paradigm

For the participant, each trial of the costly helping paradigm started with a 1 s red fixation cross, followed by a video showing the actor's reaction to the first stimulation (InputMovie, *Figure 1A*), the intensity of which was randomly determined by the computer. Participants then received an endowment of 6€, and had to decide whether to donate all or part of this money to reduce the intensity of the following stimulation. Participants knew, that every 1€ donated reduced the intensity by one unit, and that money was not donated directly to the confederate, who did not have monetary benefit from the donation. Instead, 10% of the money not donated was added as a 'bonus' to participants' compensation for taking part in the study. InputMovie intensity varied from trial to trial to cover the entire range from 2 to 7, and was randomly chosen. Participants' implicit task was to infer the intensity of the stimulation from the confederate's reaction. Participants selected the donation amount by moving a rectangle along a bar with possible donations (0€ to 6€). The starting position of the cursor was randomized to avoid motor preparation of the response. Participants were

**Table 2.** Number of videos for each intensity presented as InputMovie in the EEG and TMS experiment.
The last line shows the average movie intensities and their standard deviation presented for each condition and experiment.

| Perceived intensity | EEG experiment | | TMS experiment per session | |
| --- | --- | --- | --- | --- |
| | Hand | Face | Hand | Face |
| 2 | 13 | 23 | 4 | 10 |
| 3 | 2 | 3 | 10 | 4 |
| 4 | 0 | 4 | 4 | 4 |
| 5 | 27 | 10 | 10 | 6 |
| 6 | 1 | 11 | 2 | 6 |
| 7 | | 1 | | |
| Average intensity | 4 ± 1.4 | 3.7 ± 1.7 | 3.9 ± 1.2 | 3.8 ± 1.6 |

DOI: https://doi.org/10.7554/eLife.32740.014

instructed to use a two-pedal controller with their right foot to select the donation (USB Double Foot Switch II, Scythe Co., Ltd., Tokyo, Japan). We used a foot-pedal rather than a traditional button-box because SI activity in the hand region, critical for this experiment, may otherwise have been contaminated by the planning/performance of the button-presses. After three seconds without pressing any pedal, the software would select the current position of the rectangle as the chosen donation and move to the following black screen, which had a random duration from 1.5 to 3 s. Finally, depending on the donation, 2 s feedback video (OutputMovie) showed the confederate's response to the second stimulation. OutputMovie would represent the end of a trial. A 5 s black screen separated consecutive trials.

## EEG study

For the entire duration of the costly helping paradigm, electrophysiological brain signals were recorded from 64 active channels (10–20 positioning) by an ActiCHamp Brain Vision system. The ground electrode was placed on Fz. Electrode impedances were kept below 5 kΩ, and all signals were digitized (rate of 500 Hz) and stored for off-line processing. All data analyses were performed using the FieldTrip Toolbox (*Oostenveld et al., 2011*) and customized MATLAB (Mathworks Inc., Natick, MA, USA) scripts. The signals were low-pass filtered at 60 Hz and band-stop filtered within the range of 49.5–50.5 Hz, and harmonics were used to eliminate the electrical line noise. The data were re-referenced to a common average and segmented in epochs of 7 s, containing 5 s before InputMovie onset and lasting until InputMovie end. The segmented signals were visually inspected across all channels. Trials containing muscular and other non-ocular movement artifacts were discarded. The artifact rejection procedure resulted in 41.9 ± 1.7 artifact-free trials in the Hand task and 50.6 ± 1.9 in the Face task. Blinks and eye movements were corrected using Independent Component Analysis (*Jung et al., 2000*). Each trial was then baseline corrected using the average of the signal from 400 to 200 ms before the appearance of the fixation cross.

We used a linear constrained minimum variance beam-former approach (*Van Veen et al., 1997*). We used a three-layer BEM volume conductance model of 1 cm$^3$ resolution as the forward model (*Figure 2B*, *Oostenveld et al., 2003*). For each participant, we used the BEM model together with the covariate matrix of the ERP (entire trial length, obtained by averaging all of the videos belonging to each condition) to create an adaptive spatial filter such that its inverse applied to the sensor level representation would reconstruct the source power with maximum strength at the location of the source of interest, and would suppress the output from the sources of no-interest. To derive the complex source estimates, the time-course of each trial was multiplied with the real-valued filters. The procedure resulted in three orthogonal dipoles within each ROI, oriented in the three spatial dimensions. We constructed our filters separately for the Face and Hand sessions to account for the possibility that the noise structure changed during the long pause (~20 min) separating the two sessions. This maximized our ability to compare different trials from the same session to establish whether donation and brain activity are related within a condition, but reduced our ability to compare the two conditions directly.

To test whether brain activity in the ROIs while watching the videos could explain the Z-donation, we first conducted a mass-univariate robust regression analysis (within subjects) with the brain activity as the predictor variable and Z-donation as the observed variable. This was done separately for each time point, dipole, task, subject and ROI. Robust regression was chosen to be less sensitive to outliers in the data (*Wager et al., 2005*). The regression slopes of this subject level analysis were then subjected to Hotelling's T-Squared test (Ht2) at the group level. This is a multivariate test that examines whether the average slopes for the three dipoles are all zero. This test was repeated for each time point, condition and ROI. The family-wise error rate arising from multiple comparison of time-points was dealt with by using a cluster-based non-parametric Monte-Carlo correction. Neighboring values exceeding the cluster-cutting threshold (corresponding to Ht2 >4.675 and punc <0.01) were combined into a single cluster. Cluster-level statistics were computed by comparing the summed Ht2 values of each cluster against a permutation distribution. The permutation distribution was constructed by randomly flipping the sign of all of the slopes of randomly selected participants (1000 iterations) and by calculating the maximum group cluster statistic for each iteration. The null distribution of the cluster-based test statistic was obtained by taking the most extreme value of the statistic in each permutation. The cluster-based test statistic in the observed data was then associated with a corrected pseudo-p value based on its percentile in the null distribution for

each cluster. Furthermore, we corrected for four ROIs (two left and two right mirror ROIs) and two conditions tested using a Bonferroni correction of 6 (*Figure 3*), leading to a pseudo-p-value of 0.0063 as the cut-off for the cluster-based test statistic.

## TMS study

Sample size for the TMS experiment was determined though a power analysis conducted using G*Power 3 (*Faul et al., 2007*), with power (1 − β) set at 0.95 and α = 0.05. The effect size was chosen on the basis of the work of *Paracampo et al. (2017)* (Cohen's d = 0.94), because this work was conducted by the experimenter who was also responsible for the TMS part of the work described in the current manuscript (RP), and because the *Paracampo et al. (2017)* study involved a task that made similar cognitive demands, used an equivalent rTMS protocol, and targeted the same brain region (SI). A comparable effect size (Cohen's d = 0.89) was found when taking into account TMS studies in which participants are required to observe others and understand their behavior (*Paracampo et al., 2017, 2018*; *Valchev et al., 2017*; *Tidoni et al., 2013*). In our within subjects study, we hypothesed a perturbation of the activity in SI, and therefore a reduction in performance as a consequence of the perturbation, so we conducted a power analysis to compare performance in active stimulation with performance in sham stimulation using a matched paired one-tailed t-test at the second (group) level. This analysis yielded a required sample size of 15 participants.

TMS was administered using a figure-of-eight coil (diameter: 70 mm) connected to a Magstim Rapid2 stimulator (Magstim, Whitland, Dyfed, UK). To set rTMS intensity and to determine coil location, the resting motor threshold (rMT) was estimated for all participants in a preliminary phase of the experiment using standard procedures (*Rossi et al., 2009*). Motor-evoked potentials (MEPs) induced by stimulation of the left motor cortex were recorded from the right first dorsal interosseous (FDI) by means of a Biopac MP-35. EMG signals were band-pass filtered (30–500 Hz) and digitized (sampling rate: 5 kHz). Pairs of Ag-AgCl surface electrodes (Ø35mm) were placed in a belly-tendon montage with a ground electrode on the wrist. The intersection of the coil was placed tangentially to the scalp with the handle pointing backward and laterally at a 45° angle away from the midline. The rMT was defined as the minimal intensity of stimulator output that produces MEPs with an amplitude of at least 50 μV in the FDI with 50% probability (*Rossini et al., 2015*). After the rMT procedure, the optimal scalp location for the hand representation in the left motor cortex was marked.

A large body of evidence shows that the hand area in the somatosensory cortex can be successfully targeted by positioning the coil 1–4 cm posterior to the motor hotspot (*Harris et al., 2002*; *Balslev et al., 2004*; *Merabet et al., 2004*; *Fiorio and Haggard, 2005*; *Tegenthoff et al., 2005*; *Bufalari et al., 2007*; *Azañón and Haggard, 2009*; *Jacquet and Avenanti, 2015*; *Valchev et al., 2017*). This approach is based on the close correspondence between the motor and somatic homunculi (*Buccino et al., 2001*; *Yang et al., 1994*; *Schulz et al., 2004*; *Nakamura et al., 1998*; *Amunts and Zilles, 2015*; *Kuehn et al., 2014*). In line with this, we identified our region of interest using a two-step procedure. First, we localized the hand region in the primary motor cortex, corresponding to the optimal scalp position (OSP) for evoking MEPs in the FDI muscle. After that, we moved the coil 2 cm backward following a parasagittal plane, assuming that this displacement would not produce effects on M1. We tested this assumption directly by checking that TMS pulses applied at 105% rMT with the coil in the final target position did not elicit any detectable MEPs. To rule out any possible interference with the primary motor cortex, intensity for the rTMS was set at 90% of the resting motor threshold. Moreover, before the rTMS session, the position of the coil over the SI-L was verified by applying single pulses of TMS to ensure that no muscle activity was associated with our repetitive stimulations. While performing the costly helping paradigm, a time-locked single train of subthreshold 6 Hz rTMS (12 pulses, 2 s) was delivered (*Tidoni et al., 2013*; *Paracampo et al., 2017*), starting at the onset of the movie and thus covering its entire duration. During active rTMS blocks, the intersection of the coil was placed tangentially to the scalp directly above the scalp location of the target region with the handle pointing backward and laterally at a 45° angle away from the midline. Sham rTMS blocks were performed by tilting the coil by 90° over the same target region, to provide some scalp sensations and TMS sounds comparable to active stimulation but without inducing a current in the brain.

The general procedure of the experiment was the same as that for Experiment 1, except for changes in the number of trials and in the video presented, which were necessary to adapt the task to the TMS (sham vs active) set-up. Participants underwent a total of 60 trials for Face Condition and

60 trials for Hand condition. The conditions were presented in four blocks (two blocks for the Face and two for the Hand condition), separated by a long break in the middle in which participants further interacted with the confederate. Each block was equally divided into two parts, which were assigned to active and sham TMS in a pseudo-randomized order (e.g. for one participant: Active-TMSFaceBlock1part1, ShamTMSFaceBlock1part2, ShamHandBlock1part2, ActiveHandBlock1part2, long break, ActiveHandBlock2part1, ShamHandBlock2part2, ActiveFaceBlock2part2, ShamFace-Block2part1). Although some videos might have been shown twice within block1 or block2, videos in block1 were different from those in block2. At the end of the experimental session, participants had to rate from 1 to 4 how much headache, neck stiffness, itching on the skin, pain on the skin below the stimulation site, sleepiness and mood-swing they experienced, and whether it was difficult to concentrate. Answers were then compared with the TMS effect in the two tasks by calculating the difference in slopes between the Active and the Sham Condition and by correlating this difference with the answer to these questions across participants. Results were corrected for multiple comparison and found to be non-significant.

## HD-tDCS study

For this experiment, in addition to the Face and Hand movies, a new set of videos was created, in which no pain was depicted but the color saturation changed over time (*Figure 1B*, and *Videos 5–6*). To match the temporal dynamic of the Hand and Face videos, the saturation change started after 1 s from the beginning of the videos and reached its peak 0.5 s afterwards. Videos were created with three different levels of saturation changes. An independent group of 20 participants (*Table 1*) watched Color, Face and Hand videos (presented using EventIDE; OkazoLab Ltd., 2012), and rated from 1 to 10 how painful the stimulation was for the person in the Face and Hand videos, and how much the saturation changed in the Color videos. Using their left hand, they moved a rectangle along a bar with possible ratings, from 0 to 10. The starting position of the cursor was randomized to avoid motor preparation of the response. The validation procedure resulted in a total of 32 videos per category matched for average rating ($F_{(2,93)}$=0.2, p=0.8) and accuracy, calculated as the square of the difference from the expected value ($F_{(2,93)=}$0.4, p=0.6).

Sample size for the tDCS experiment was determined though a power analysis conducted using G*Power 3 (*Faul et al., 2007*), with power (1 – β) set at 0.95 and α = 0.05. We expected a small effect size on the basis of recent transcranial electrical stimulation experiments (*Bolognini et al., 2013*; *Avenanti et al., 20172018*). In these studies, the somatosensory cortices were targeted, and similar design and task requirements were used. This analysis yielded a required sample size of 26 participants (*Table 1*).

1.5 mA was delivered to the left primary somatosensory cortex for 18 min through a 4 × 1 ring-electrode set-up consisting of a central active anode and four surrounding return electrodes (*Kuo et al., 2013*; *Figure 2D*), which were connected to a battery-operated tDCS MXN-9 High-Definition (HD) Stimulator (Soterix Medical Inc., USA). The HD-tDCS electrodes were fixed to a cap by means of HD-tDCS electrode holders, with the central anode placed over the primary somatosensory cortex, between the EEG electrode sites C3 and CP3. The HD-tDCS electrodes' impedance was kept below 10 kΩ. Throughout the stimulation, the participant was sitting comfortably in an arm chair.

Participants received both real and sham stimulations on two different days, separated by an average of 7.8 days (SD = 3). After the sham and real stimulations, participants performed the rating task as in the validation procedure. Each trial started with the presentation of a white fixation cross (1 s), which was followed by the video clip (2 s), and finally the visual analogue scale. Using their left hand, participants moved a rectangle along the scale using two keys (one for moving the rectangle to the right, one to the left). A third key was pressed to confirm the intensity selection. A variable interval of between 2800 and 3200 ms separated the trials. Videos were presented in six blocks (two per task) containing multiple intensities. Block presentation order was randomized among participants and among sessions.

After both the real and the sham sessions, participants rated from 1 to 4 how much headache, neck stiffness, itching on the skin, pain on the skin below the stimulation site, sleepiness and mood-swing they experienced, and whether it was difficult to concentrate.

# Additional information

## Funding

| Funder | Grant reference number | Author |
|--------|------------------------|--------|
| Nederlandse Organisatie voor Wetenschappelijk Onderzoek | VIDI: 452-14-015 | Valeria Gazzola |
| Brain and Behavior Research Foundation | NARSAD young investigator 22453 | Valeria Gazzola |
| H2020 European Research Council | ERC-StG-312511 | Christian Keysers |
| Cogito Foundation | R-117/13 | Alessio Avenanti |
| Fundação Bial | 298/16 | Alessio Avenanti |
| Cogito Foundation | 14-139-R | Alessio Avenanti |
| Nederlandse Organisatie voor Wetenschappelijk Onderzoek | VICI:453-15-009 | Christian Keysers |

The funders had no role in study design, data collection and interpretation, or the decision to submit the work for publication.

## Author contributions

Selene Gallo, Conceptualization, Data curation, Formal analysis, Supervision, Validation, Investigation, Visualization, Methodology, Writing—original draft, Project administration, Writing—review and editing; Riccardo Paracampo, Supervision, Validation, Methodology, Writing—review and editing; Laura Müller-Pinzler, Tatjana Maskaljunas, Formal analysis, Investigation, Writing—review and editing; Mario Carlo Severo, Formal analysis, Validation, Investigation, Writing—review and editing; Laila Blömer, Validation, Investigation, Writing—review and editing; Carolina Fernandes-Henriques, Anna Henschel, Balint Kalista Lammes, Judith Suttrup, Investigation, Writing—review and editing; Alessio Avenanti, Funding acquisition, Methodology, Writing—review and editing; Christian Keysers, Conceptualization, Resources, Supervision, Funding acquisition, Methodology, Writing—original draft, Writing—review and editing; Valeria Gazzola, Conceptualization, Resources, Data curation, Supervision, Funding acquisition, Visualization, Methodology, Writing—original draft, Project administration, Writing—review and editing

## Author ORCIDs

Selene Gallo ⓘD http://orcid.org/0000-0003-4152-8808
Riccardo Paracampo ⓘD http://orcid.org/0000-0001-9399-1306
Mario Carlo Severo ⓘD https://orcid.org/0000-0001-7403-819X
Carolina Fernandes-Henriques ⓘD http://orcid.org/0000-0002-4019-2049
Anna Henschel ⓘD http://orcid.org/0000-0003-1094-1587
Judith Suttrup ⓘD http://orcid.org/0000-0002-4034-1534
Alessio Avenanti ⓘD http://orcid.org/0000-0003-1139-9996
Christian Keysers ⓘD http://orcid.org/0000-0002-2845-5467
Valeria Gazzola ⓘD http://orcid.org/0000-0003-0324-0619

## Ethics

Human subjects: All studies have been approved by the Ethics Committee of the University of Amsterdam, the Netherlands. Project identifiers: 2016-BC-7394 2016-BC-7130 2016-PSY-6485 2014-EXT-3476 2014-EXT-3432 All participants received monetary compensation and gave their informed consent for participation in the study.

## Decision letter and Author response

Decision letter https://doi.org/10.7554/eLife.32740.038
Author response https://doi.org/10.7554/eLife.32740.039

## Additional files

### Supplementary files

• Supplementary file 1. Source data: for each of the dipoles extracted from d-SIL, the single participant's average brain activity within the time-window was able to predict their donation (significant in cluster-statistic).
DOI: https://doi.org/10.7554/eLife.32740.024

• Supplementary file 2. Source data: for each of the dipoles extracted from v-SIL, the single participant's average brain activity within the time-window was able to predict their donation (significant in cluster-statistic).
DOI: https://doi.org/10.7554/eLife.32740.025

• Supplementary file 3. Source data: data used for gender difference analyses.
DOI: https://doi.org/10.7554/eLife.32740.026

• Source code 1. Pain Localizer source code.
DOI: https://doi.org/10.7554/eLife.32740.027

• Source code 2. Pain Localizer.
DOI: https://doi.org/10.7554/eLife.32740.028

• Transparent reporting form
DOI: https://doi.org/10.7554/eLife.32740.029

### Data availability

fMRI and EEG data have been deposited in Zenodo. Source data files have been provided for all figures.

The following datasets were generated:

| Author(s) | Year | Dataset title | Dataset URL | Database, license, and accessibility information |
|---|---|---|---|---|
| Gallo S, Paracampo R, Müller-Pinzler L, Severo MC, Blömer L, Fernandes-Henriques C, Henschel A, Lammes BK, Maskaljunas T, Suttrup J, Avenanti A, Keysers C, Gazzola V | 2018 | The causal role of the somatosensory cortex in prosocial behavior - Pain Localizer | https://doi.org/10.5281/zenodo.1213175 | Publicly available at Zenodo (https://zenodo.org/) |
| Gallo S, Paracampo R, Müller-Pinzler L, Severo MC, Blömer L, Fernandes-Henriques C, Henschel A, Lammes BK, Maskaljunas T, Suttrup J, Avenanti A, Keysers C, Gazzola V | 2018 | The causal role of the somatosensory cortex in prosocial behavior - EEG dataset | https://doi.org/10.5281/zenodo.1213584 | Publicly available at Zenodo (https://zenodo.org/) |

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

# Appendix 1

DOI: https://doi.org/10.7554/eLife.32740.030

## Supplementary information

### Pain localizer and region of interest definition

An fMRI experiment (Pain Localizer) was performed to identify areas of the somatosensory cortex that encode the subjective intensity of pain stimulation. These regions were then interrogated via beamforming during the costly helping paradigm. Twenty-five participants (age mean = 24, SD = 5.6, nine males) with no reported neurological, psychiatric or other medical problems, and no contraindication to fMRI, participated in the experiment. No discomfort was reported by participants or noticed by the experimenter. All participants were paid for their participation. The study has been approved by the Ethics Committee of the University of Amsterdam, The Netherlands. Participants underwent a total of 40 electrical and 40 mechanical stimulations, split into eight runs (four electrical and four mechanical) of ten (five high intensity and five low intensity) stimulations each, while BOLD changes were recorded. High and low stimulations were delivered in a pseudo-randomized order (with no more than two consequential stimulations of the same intensity) on the participant's right hand using a MRI-compatible stimulation system (DS7A stimulator - Digitimer Ltd, Welwyn Garden City, Hertfordshire, UK). After a random interval ranging from 2 to 5 s, a question mark appeared on the screen and participants were asked to report their pain intensity by selecting, with their left hand, 1 of 4 available buttons. Each button corresponded to a double step on a scale from 1 to 10, in which one corresponded to a non-painful stimulation and 10 the most intense imaginable pain. Because we were not allowed to deliver intensity higher than a reported value of 8, the fifth button (9–10) was not given as a possible option (1–2 index finger, 3–4 middle finger, 5–6 ring finger, 7–8 little finger). The question mark disappeared as soon as a button was pressed. A variable interval of 8–12 s separated the rating period from the next stimulation. Electrical stimulations lasted 0.5 s and current intensity was delivered and chosen for each participant as described in the description of the video recording procedure. We selected current intensity consistently rated as 2 and 6 prior to scanning, and used these intensities as low and high stimulation currents during data collection. The mechanical stimulation was delivered through a rubber band mounted on a plastic tube cut longitudinally: the participant's hand was resting inside the tube and the experimenter would snap the rubber band on the participant's hand following auditory instructions. Also in this case, participants underwent a threshold-like procedure similar to that described for the electrical stimulation.

A Phillips Achieva 3.0 T MRI scanner was used to acquire T2*-weighted echo-planar (32 interleaved 3.5 mm thick axial slices, 0.35 mm gap, TR = 1700 ms, TE = 27.6 ms, flip angle = 73°, FOV = 240 mm×240 mm, 80 × 80 matrix of 3.5 mm isotropic voxels), and whole brain T1-weighted anatomical (1 × 1 × 1 mm) images. Each stimulation block lasted a minimum of 3.8 min to a maximum of 6.5 min. FMRI data was pre-processed using SPM12 (www.fil.ion.ucl.ac.uk). All echo planar images (EPIs) were slice-time corrected and realigned to the participant's mean EPI. The T1 image was co-registered to the mean EPI and segmented. The estimated normalization parameters were applied to all EPIs. Normalized (2 × 2 × 2 mm) EPIs were smoothed with a 6 mm isotropic FWHM Gaussian kernel. A general linear model (GLM) was applied at the single-subject level. Predictors were modelled using a standard boxcar function convolved with the hemodynamic response function (HRF). For each of the eight runs of the pain localizer, we included: a 5 s-length predictor modelling the instruction screen at the beginning of each run, a 500 ms predictor aligned to the electrical stimulations, a 200 ms predictor aligned to the mechanical stimulations, and a predictor aligned to the appearance of the question mark until the participant's button response. For both of the predictors aligned to the stimulations, the participant's rating was included as a linear parametric modulator. If a participant pressed

two or more different buttons after the stimulation, it was not possible to assign a subjective painfulness to the event and so this stimulation was excluded the main predictor. Such events were included in a separated predictor of no interest, as were button presses outside the rating period. Six additional regressors of no interest were included to account for head movements (none of the included participants had frame-wise displacement along x, y or z exceeding the acquired voxel-size). The modulators of the mechanical and electrical stimulation were added and this sum was tested against zero at the second level by means of a t-test. The contrast identified a number of clusters surviving voxelwise FDR at $q < 0.05$ ($t > 2.23$). Because the uncorrected threshold at $p<0.001$ led to more conservative t values ($t > 3.5$), we used this threshold to draw our ROIs. ROIs were defined by calculating (ImCalc) the voxels common to the cytoarchitectonic maximum probability maps of SI Left (BA 3a, 3b, 1 and 2) from the Anatomy toolbox for SPM (*Eickhoff et al., 2005*) and were modulated by perceived pain intensity ($t > 3.5$). This analysis revealed a dorsal (dSI-L) and a ventral (vSI-L) cluster. Given the spatial resolution of the forward model used in the EEG beamforming (1 cm), we rounded the MNI coordinates of the activation peak of each cluster (dSI-L X,Y,Z = −30,–40, 60 and vSI-L X,Y, Z = −50,–20, 30) to guide the EEG source reconstruction procedure (*Figure 2* main text). Mirror ROIs were created by flipping sign of the coordinate of the Y axis (dSI-R X,Y,Z = −30, 40, 60 and vSI-R X,Y, Z = −50, 20, 30).

## EEG experiment: mirror ROIs results

We replicated the procedure described in the main test to test whether brain activity originating from the mirror ROIs (dSI-R and vSI-R) also predicts donation. Results show that activity in dSI-R predicts Z-donation in the Hand condition in the time window between 402 and 484 ms (cluster-statistic = 318.5, $p<0.0001$) after the belt hits the confederate's hand and predicts Z-donation in Face between 522 and 562 ms (cluster-statistic = 161.0, $p<0.005$). vSI-R brain activity predicted Z-donation but only in the Face condition in the time window between 444 and 676 ms (cluster-statistic = 804.3, $p<0.0001$), and between 706 and 738 ms (cluster-statistic = 804.3, $p<0.0063$; *Figure 3C*). Results survive multiple comparisons.

## Correlation with self-reported questionnaires

At debriefing, participants completed the Interpersonal Reactivity Index questionnaire (IRI, *Davis, 1983*) and the Money Attitude Scale (MAS, *Yamauchi and Templer, 1982*) to measure empathy related traits and attitude towards money, respectively.

To assess whether these personality traits explained the participants' differences in average donation we performed, in the costly helping experiment with the highest power (EEG, 28 participants) a multiple regression with the four IRI subscales (Empathic Concern, Perspective Taking, Fantasy Scale and Personal distress), and the MAS score as predictor of the average donation. In line with the notion that average donation reflects a trait, individuals that gave high average donations in the Face condition also gave high average donations in the Hand condition (r(Hand,Face)=0.85). We therefore used the grand average donation across both conditions as the dependent measure in the regression. The regression that resulted was significant ($F(5,22)=3.5$, $p=0.02$, $r2 = 0.45$). In particular, Empathic Concern and the MAS significantly explained the average donation: the more participants rated themselves as having high empathic concern towards others and the less they affirmed to value money, the more they donated during the Costly Helping Paradigm (EC: b = 0.4, $p=0.02$; MAS: b = −0.5, $p=0.02$; all others $p>0.4$).

Examining individual differences in the slope that links the intensity of pain in the InputMovie (as judged by an independent sample) with the trial-by-trial donation led to very different results. First, the slopes in the Face and Hand condition were not strongly correlated (r(Hand,Face)=0.23), and multiple regressions performed separately for the Face and Hand slopes revealed that none of the scales significantly explained variance in the slopes (all $p>0.1$).

The results described above are not replicated within the smaller group of participants for the TMS study (N = 15) and should therefore be considered tentative.

*Supplementary file 1*. Hand videos trials have been classified as low- and high-donation (median split per participant). The table indicates, for each of the dipoles extracted from d-SIL, the average amplitude and standard deviation within time-window able to predict donation (significant in cluster-statistic), and the comparison between low- and high-donation trials calculated using a paired t-test. Significant results are highlighted.

*Supplementary file 2*. Face-videos trials have been classified as low- and high-donation (median split per participant). The table indicates, for each of the dipoles extracted from d-SIL and v-SIL, the average amplitude and standard deviation within time-window able to predict donation (significant in cluster-statistic), and the comparison between low- and high-donation trials calculated using pair t-test. Significant results are highlighted.

## Exploring gender differences

There is some evidence in the literature that shows gender differences in empathic behavior in a broad range of measures, not only in humans but also in other animals (see *Christov-Moore et al. [2014]* for a recent overview). Particularly relevant for the present studies are the differences regarding face processing. Women are more accurate and/or efficient in processing facial expressions of emotions in general (*Hoffman, 1977*; *Hall and Matsumoto, 2004*; *Korb et al., 2015*; *Proverbio, 2017*) and of pain in particular (*Keogh, 2014*). Women show more facial mimicry than men (*Dimberg and Lundquist, 1990*), and are also better than men at recognizing pain when it is expressed by the body posture (*Walsh et al., 2017*). In this section, we therefore explored the effect of gender in our data. Given the low number of participants of each gender, these analyses should be interpreted in the light of their low statistical power. Detailed analyses are presented below, but in summary, we found no significant evidence for gender effects in either our EEG studyor in our neuromodulation studies.

### Experiment 1: EEG study (15 males, 13 females)

Behaviorally, there was no evidence for gender differences in average donation or slopes: Hand Condition average donation — males mean = 2, SD = 1.2; females mean = 2.3, SD = 1.2; independent groups t-test $t(26)=0.7$, $p=0.4$; Face Condition average donation — males mean = 2, SD = 1.3, females mean = 2.2, SD = 1.2; independent groups t-test $t(26)=0.4$, $p=0.7$; Hand Condition average slope — males mean = 0.5, SD = 0.1; females mean = 0.4, SD = 0.2; independent groups t-test $t(26)= -1.2$, $p=0.2$; Face Condition average slope — males mean = 0.5, SD = 0.05; females mean = 0.5, SD = 0.07; independent groups t-test $t(26) = -1.7$, $p=0.1$.

To explore potential gender effects in the relationship between given donation and brain activity recorded in dSI-L and vSI-L, we first averaged together the slopes belonging to all the significant time-points in each task. Then using a t-test for independent groups for each cluster, we compared the average in the genders. *Supplementary file 3* summarized the results. There is no evidence for gender difference in any of the cluster averages.

*Supplementary file 3*. Mean and standard deviation of the average of all the significant time-points belonging to a cluster, divided for tasks, ROI and dipole orientation and gender. t-values and p-values of the t-tests for independent groups (gender) are reported.

### Experiment 2: TMS study (nine males, six females)

We performed the same analysis on the slopes described in the main text adding 'gender' as a group factor. This new mixed ANOVA with two repeated factors (Condition and TMS) and an among participants factor (Gender) did not reveal any gender effect: main effect of Gender $F(1,13)=0.008$, $p=0.9$; Condition X Gender interaction $F(1,13)=0.02$, $p=0.9$; TMS X Gender interaction $F(1,13)= 0.9$, $p=0.3$; Condition X TMS X Gender interaction $F(1,13)= 0.06$, $p=0.8$.

### Experiment 3: HD-tDCS study (13 males, 13 females)

As in experiment 2, we repeated the ANOVA described in the main text adding 'Gender' as a group factor. This new mixed ANOVA with two repeated factors (Stimulation and Condition) and an among participants factor (Gender) showed no significant interaction with between Gender and Condition or Stimulation factors (Interaction Stimulation X Gender $F_{(1,23)}$=0.3, p=0.6; Condition X Gender $F_{(1,23)}$=1.5, p=0.2; Stimulation X Condition X Gender $F_{(1,23)}$=0.03, p=1), suggesting that the effect of our experimental manipulation does not depend on gender. However, the ANOVA shows a main effect of Gender ($F_{(1,23)}$=10, p=0.004): females had on average a steeper slope then males in all the tasks, independently of Condition and Stimulation. This result does not seem to be related to differences in empathic ability, because in the control task females also showed a steeper slope. Instead, the result might reflect a general and unspecific difference in involvement in the task.

## Correlation with self-reported questionnaires

Participants from Experiment 1: EEG study completed the Interpersonal Reactivity Index questionnaire (IRI, *Davis, 1983*) and the Money Attitude Scale (MAS, *Yamauchi and Templer, 1982*) to measure empathy-related traits and attitude towards money, respectively.

To assess whether these personality traits explained participants' differences in performance in the Costly Helping Paradigm, we performed a multiple regression with the four IRI subscales (Empathic Concern, Perspective Taking, Fantasy Scale and Personal distress) and the MAS score as predictor of the average donation given by the participants on average to the Face and Hand conditions. For both conditions, the regression was significant (Face: $F_{(5,22)}$=3.8, p=0.01, $r^2$ = 0.5; Hand: $F_{(5,22)}$=2.6, p=0.05, $r^2$ = 0.4). In particular, Empathic Concern and the MAS were significant predictors of the average donation both in the Face and the Hand condition: the more participants rated themselves as having high empathic concern towards othesr and the less they affirmed to value money, the more they donated during the Costly Helping Paradigm (Face-EC: b = 0.4, p=0.03; Face-MAS: b = −0.5, p=0.02; Hand-EC: b = 0.4, p=0.3; Hand-MAS: b = −0.4, p=0.03).

