## [Decision Letter]

Thank you for submitting your article "The role of the somatosensory cortex in prosocial behaviour" for consideration by *eLife*. Your article has been reviewed by four peer reviewers, one of whom Peggy Mason (Reviewer #1) is a member of our Board of Reviewing Editors and the evaluation has been overseen by Sabine Kastner as the Senior Editor. The following individuals involved in review of your submission have agreed to reveal their identity: Claus Lamm (Reviewer #2); Stephanie Preston (Reviewer #4).

The reviewers have discussed the reviews with one another and the Reviewing Editor has drafted this decision to help you prepare a revised submission.

This is a very interesting study that looked at the role of S1 activation in prosocial behavior evoked by videos suggestive of another's pain. The videos featured either a confederate's (confederacy was not known to participants) face or hand as the hand was injured. There are two primary findings. The magnitude of activation in the hand, but not face, region correlated with the amount of money given to ameliorate the confederate's future pain experience. Second, TMS in the hand region (face TMS was not tested) reduced the dependence of money donated on stimulation intensity, but only in the hand and not in the face condition. The authors interpret this as evidence that evidence that S1 activation influences prosocial decision-making. Specifically, they conclude that the intensity perceived through S1 hand activation is fed onto a decision-making area. However, this only occurs with activation that involves the body part directly affected and not S1 activation of the face.

The TMS data in particular are new and add to our understanding. The reviewers' concerns with the manuscript, were primarily rhetorical beyond the desire for experiments employing TMS of the face region, which appears to be technically very difficult. The primary revisions that the reviewers want to see are:

- clean up the language and be far more careful (=accurate where there is positive evidence – hand S1 – and where there is no evidence- face S1) about the interpretation of the data

- address power issues outlined by the reviewers

- address the scholarship issues by inclusively and accurately citing past work and theoretical framework

*Reviewer #1:*

This is a very interesting study that looked at the role of S1 activation in prosocial behavior evoked by videos suggestive of another's pain. The videos featured either a confederate's (confederacy was not known to participants) face or hand as the hand was injured. There are two primary findings. The magnitude of activation in the hand, but not face, region correlated with the amount of money given to ameliorate the confederate's future pain experience. Second, TMS in the hand region (face TMS was not tested) reduced the dependence of money donated on stimulation intensity, but only in the hand and not in the face condition. The authors interpret this as evidence that evidence that S1 activation influences prosocial decision-making. Specifically, they conclude that the intensity perceived through S1 hand activation is fed onto a decision-making area. However, this only occurs with activation that involves the body part directly affected and not S1 activation of the face.

In general, I like this study. My concerns are the following

The authors did not test TMS of the face and so they cannot speak to whether S1 face area activation influences prosocial giving. They come close to this, many times and cross the line at least once: (When pain is expressed by the face, SI activation predicts but does not influence 204 decision-making, suggesting that SI activation is then an epiphenomenon that does not influence decision making. – This is simply not supported by the data as it was not tested whether TMS activation in the face S-s area altered donations.)

The results are difficult to read and understand for someone that does not employ the methods used. They should be simplified and expanded as needed to bring along the interested non-specialist.

Just to play devil's advocate, what was the alternative to this finding? We know that the brain, including S1, codes for intensity and that more giving happens with greater viewed perceived intensity of pain. So, of course, this low level sensory intensity information from S1 is going to correlate with giving. The new thing here is that this does not work for people viewing facial expressions, a result that harkens back to a known fact introduced in the Intro [Witnessing somebody in pain activates two networks depending on the nature of the stimulus – abstract or sensory]. This disconnect should be further explored.

The sex ratio is quite different in the TMS study (2/3 male) than in the other cohorts studied (about even). Can the authors provide evidence as to whether there are any sex differences that may confound the findings given the different cohort makeups?

Figure 2C is not referred to. There is no Figure 4 but Figure 5 – which does not exist – is referred to. It appears that Figure 5 should be 2C. Obviously fix and double check all figure references.

*Reviewer #2:*

I think this is an excellent paper that would be a very good fit for a high-level journal like *eLife*. Particular strength are the multi-method approach and the converging evidence across two or rather even three experiments (see below), as well as the attempt to gain more causal evidence on the neural bases of "empathy"/vicarious neural response. I have only two issues to discuss/comment on:

1) What do the results tell us about the nature of the activations in S1?

I agree that approaches providing more causal insights are needed to advance the social neuroscience of affect and cognition. However, what remains to be shown in many cases, including my own research into this direction, is what the "causally involved" neural processes really mean in terms of "psychological/functional processes". This is why the third experiment using tDCS showing one possible mechanism – "perception of pain intensity" – is crucial, and it is currently way "undersold" in the paper/mentioned somewhat only in passing. I would incorporate this experiment into the write up of the paper from the beginning, including in the intro. And I would like to see a more extensive discussion of it and its implications. Moreover, this experiment tells us something about a possible shortcoming of the findings/an avenue for future research. The current approach basically only zooms in on activation in S1 in response to seeing hand movements in response to pain. It does unfortunately not exploit the opportunity – probably as this is much harder to implement due to the location of the brain areas involved – to also measure or inhibit activation in areas involved in face processing/facial affect discrimination. In an ideal world, the authors could aim for a double dissociation of areas involved in coding vicarious affect from observing hands vs. faces, and link this to differences in pain intensity decoding, and how they are related to prosocial action. This could be discussed more extensively as both a limitation and a future research avenue.

2) Power analysis

I am truly surprised that the power analysis for the TMS experiment yielded a sample size of N=15, especially as the targeted power is unusually high with.95. It also doesn't sound very convincing that the effect size was based "on previous related findings", as the cited paper looked at TMS effects to decoding amusement from smiles. This sounds, sorry to be so direct, somewhat suspicious. It is also unclear for what kind of statistical tests the power analysis was computed? For regression analysis? t-tests?

3) Face vs hand – sensitivity?

The authors seem to suggest based on the TMS effects that the face condition does not causally involve S1 while the hand condition does. This is an unnecessary "binary" view on things, motivated by statistics showing effects only in one case and not the other. My take of the results would rather be that the effects for the hand are stronger, while it is unclear whether sensitivity (power? see above) of the study might not have been high enough for the face condition. I am saying this (also) because the face conditions showed, consistently, higher variability in donations. This might have driven the lack of detection effects also for this condition. I would thus tone down too "binary" interpretations of the TMS findings.

*Reviewer #3:*

The current study sought to examine the role of the somatosensory cortex in prosocial behavior. The first experiment was an EEG experiment that examined if activity of SI hand representation predicts trial-by-trial donations. The second TMS experiment examined if inhibiting SI will affect donation.

The topic of the study is interesting, and the experiments are well planned.

The link between empathy and prosocial behavior is not clear in this study. It has been suggested that empathic concern rather than personal distress contributes to prosocial behavior. This issue is not addressed in the study.

The role of the SI in prosocial behavior is not properly justified in the Introduction.

There are no clear predictions in the presented in the Introduction.

The role of the face condition as presented in the Introduction is not clear

Materials and methods section:

It is not clear why the authors focused on the left SI and not the right. Other studies suggest that the right S1 is involved in empathy and there is also evidence for bilateral involvement.

TMS experiment:

Isn't it problematic to perform the task twice?

It is important to examine the effect of order of the stimulation.

How accurate was the location of S1 and why no navigation method was applied?

It seems that testing if TMS interferes with the relationship between intensity and donation is post-hoc. This analysis is not discussed in the Introduction and not justified.

*Reviewer #4:*

This is an interesting paper on an important topic.

The field very much needs research on the role of the somatosensory cortex in empathy. I appreciate the multi methods used here.

I also appreciated the places in the text where the authors gave methodological details and limitations of existing studies, which are more detailed (necessarily so) than most articles and that will contribute to people's understanding of this issue.

There are some statements made in the intro that I think are oversimplifications that could be improved. I think the existing knowledge about prosocial behavior, about the role of empathy in altruism, and about existing neuroscience tools besides TMS to study this problem are underestimated. There is abundant evidence that empathy can promote altruism but also that you can get helping for other reasons (display, status, self-interest) and that sometimes people feel empathy but don't help (because of fear, conflict of interest, distress etc.). There are also ways that neuroscience can and already has addressed some of these issues. The addition of TMS is nice but it is not necessary to make it seem like the whole field to this point has been a morass of confusion in order to justify this work. The very need to shore up some issues with our understanding of the role of this region is plenty of motivation to perform and publish this work.

Note that there is a major mis-step in the discussion where it says multiple times that because the face TMS did not alter donations that means that SI activation does not influence decision making. This is incorrect. What you have shown is that it is not necessary for or fully explanatory of decision making in the current context. There are multiple reasons to not make the assumption that you did. For example, people are expert facial readers who can make explicit and top-down inferences about internal states from expressions without having to rely on neural/physiological feeling states once they are "trained up" in an embodied system and learn cultural display rules. In addition, people can use that information normally as inputs to the decision even if they can find other ways (like the aforementioned) to make the choice when that input is not available.

I am concerned about the distribution of pain intensity in Table 2. The TMS face condition has a skewed distribution toward the lowest intensity and does not have the second peak of higher intensity that is available for the other three conditions. I think you have to assume that any differences in the Face condition, especially null effects in the TMS study could reflect small(er) effect sizes because there is not enough high intensity information to drive effects.

Note that the Preston and de Waal, 2002 paper is cited at the beginning as evidence that there is a lot of confusion over empathy producing altruism. That is not an accurate depiction of the contents of that paper. It is also not included in the reference section. The fact that the 2002 paper was the first systematic review to even suggest that empathy relied on overlapping neural substrates between felt and observed pain is overlooked and should be cited. That theory was even based on evidence from Adolphs and colleagues showing that somatosensory cortex per se is needed to decode facial emotion--a study that used lesion patients (and, thus, has good causal inference power, in conflict with claims here that this is the first time ever that one could make claims about causality with this region in emotion perception or empathy…). I think this citation issue and credit needs to be addressed. You can also see a new review, de Waal and Preston, 2017, for an updated version that clearly includes somatosensory in the map of empathy-related areas.

I'm not sure you can conclude that SI activity predicting the donation in the second half means that only the reaction to the pain is relevant since the response to both (belt hitting hand and person reacting) could be used and have to summate or build with additional processing to reach the threshold necessary to explain donations. Also, the belt video I saw does not have a very fast, salient, painful looking hit, so I would assume that the NS for that segment can easily be an effect size problem with the stimuli that doesn't look super painful.

---

## [Author Response]

The TMS data in particular are new and add to our understanding. The reviewers' concerns with the ms were primarily rhetorical beyond the desire for experiments employing TMS of the face region, which appears to be technically very difficult. The primary revisions that the reviewers want to see are:- clean up the language and be far more careful (=accurate where there is positive evidence – hand S1 – and where there is no evidence- face S1) about the interpretation of the data.

We have edited the manuscript accordingly.

- address power issues outlined by the reviewers.

We now include Bayesian analyses in cases of non-significant results to explore whether the non-significance supports the null-hypothesis or whether the non-significant results simply reflects a lack of power.

- address the scholarship issues by inclusively and accurately citing past work and theoretical framework.

We now include the papers suggested by the reviewers and refined out presentation of the theoretical framework.

Reviewer #1:

[…] The authors did not test TMS of the face and so they cannot speak to whether S1 face area activation influences prosocial giving. They come close to this, many times and cross the line at least once: (When pain is expressed by the face, SI activation predicts but does not influence 204 decision-making, suggesting that SI activation is then an epiphenomenon that does not influence decision making. This is simply not supported by the data as it was not tested whether TMS activation in the face S-s area altered donations.)

We thank the reviewer for this important comment that points to two issues. First, we had failed to rigorously specify in the text that our statements apply to the hand region of SI, rather than to SI in general. We went through the manuscript and reformulated ambiguous sentences accordingly. Second, our core result is an interaction between TMS condition (sham vs. active) and Stimulus condition (Face vs. Hand) on the regression slope. This provides evidence that the SI hand region is more necessary in determining donation based on information derived from the Hand than for the Face stimuli. We thus revised statements suggesting that we have evidence that the Hand region is not necessary in determining donation from the Face information to statements specifying that we have evidence that it is less involved.

Moreover, to address whether the non-significant effect of disrupting the SI hand region with TMS while viewing Face stimuli is due to a small sample size or is indicative of the absence of a sizable effect, we complement the more traditional frequentist analysis with a Bayesian statistic.

In subsection “Experiment 2: TMS study*”* we write: “To test if the lack of effect in the Face condition was due to limited statistical power or provides evidence for the null hypothesis (H_0_) of no (sizable) effect of TMS, we used Bayesian statistics. We calculated an index of TMS effect for each participant in the Face condition as follows: slope in the Active session – slope in the Sham session. We then performed a Bayesian one sample t-test using JASP with default priors (https://jasp-stats.org) that showed the null hypothesis is 5.7 times more likely than the alternative H_1_ (Bayes factor (H_0_:index≥0|data)/p(H_1_:index<0|data)=5.7) providing positive evidence for the absence of a sizable effect in the Face condition (Kass and Raftery, 1995)”.

In subsection “Experiment 3: HD-tDCS study*”*: “Since Hand and Face conditions showed opposite effect, we performed on the Color condition a 2-tails one sample Bayesian t-test. The Bayes factor indicated that the H_0_ was 3.1 times more likely than the H_1_, confirming that the Color condition does not change after HD-tDCS. The Hand and Face effects did not correlate with any of the tDCS side effect perceived by participants while performing the experiment (all p>0.05).”

As mentioned above, throughout the text we now focus on the interpretation derived from the significant interaction (i.e. the effect is smaller in the Face compared to the Hand condition). Additionally, in the discussion we now mention more explicitly, that this lesser involvement may reflect multiple scenarios:

“When pain is expressed by the reaction of the hand, the dorsal SI hand region activation correlates with decision making (EEG). Additionally, altering this activity changes (TMS) decision making, suggesting that the SI hand region activation feeds into the decision-making process. When pain is expressed by the face, the SI hand region activation correlates with decision making (EEG) but altering this activity influences decision making less. This latter finding is compatible with three interpretations. (i) Activity in the hand-region of SI is an epiphenomena, representing an imagination of what the painful stimulation would have felt on the hand (Fairhurst et al., 2012), that is not used in decision making. (ii) The SI hand region activity is used for decision making in the sham condition but can be substituted by alternative sources of information derived from the face elsewhere in the brain in the active TMS condition. Both these interpretations are reminiscent of the notion that pain information takes different paths based on the stimulus it is derived from (Keysers, Kaas and Gazzola, 2010; Lamm, Decety and Singer, 2011). (iii) Information in the SI hand region has a higher signal to noise ratio in the Face condition compared to the Hand condition, making it less susceptible to TMS interference.”

The results are difficult to read and understand for someone that does not employ the methods used. They should be simplified and expanded as needed to bring along the interested non-specialist.

We worked on the Results section accordingly to the reviewer’s suggestion, hoping to have reached a level of simplification accessible to more readers. In particular we expanded the rationale behind the analyses and we tried to provide more accessible explanations for each step.

Just to play devil's advocate, what was the alternative to this finding? We know that the brain, including S1, codes for intensity and that more giving happens with greater viewed perceived intensity of pain. So, of course, this low level sensory intensity information from S1 is going to correlate with giving. The new thing here is that this does not work for people viewing facial expressions, a result that harkens back to a known fact introduced in the Introduction (Witnessing somebody in pain activates two networks depending on the nature of the stimulus – abstract or sensory). This disconnect should be further explored.

Although not all neuroscientist would agree that SI codes the intensity of perceived pain, similarly to the reviewer, we predicted such a relationship for the hand stimuli, and agree that this disconnect should be further investigated. In the Discussion section we now write:

“Future research should neuro-modulate brain activity in ventral SI in addition to the hand representation we targeted here while measuring the willingness to help in order to further investigate the dissociation suggested by our results. The emergence of focused ultrasounds as a focal neuro-modulation method (Mueller et al., 2014; Lee et al., 2015; Lee et al., 2016),could enable such studies without the muscle artefacts inevitable with TMS. HD-tDCS, as used in our third experiment, also has the advantage not to cause muscle twitches, but lacks the focality to argue with confidence that one can disentangle the contribution of the face and hand region located only 2cm away.”

And in the concluding paragraph: “These neuromodulation findings support the notion derived from neuroimaging literature that multiple networks can be recruited during the perception of the pain of others depending on the nature of the stimulus(Keysers, Kaas and Gazzola, 2010; Lamm, Decety and Singer, 2011). Future studies will be needed to isolate and characterize the causal contribution and interaction across the nodes of these networks, and further characterize the conditions under which each network is necessary.”

The sex ratio is quite different in the TMS study (2/3 male) than in the other cohorts studied (about even). Can the authors provide evidence as to whether there are any sex differences that may confound the findings given the different cohort makeups?

We performed supplementary analyses to assess gender effects. For gender effects to have confounded the quantification of the effects of Condition (Face vs Hand) or Stimulation (Active vs. Sham), there must have been a significant interaction of Gender with either or both of these factors. Our supplementary analyses rule this out: in all of the studies gender did not interact with Condition or Stimulation. All we found was a single main effect of gender in the HD-tDCS study, with women having on average steeper slopes. To provide information for readers interested in gender effects, despite the underpowered nature of these analysis in our study, we now include a table with all gender effects in subsection “Exploring Gender Effects”).

Exploring gender differences

There is some evidence in the literature for gender differences in empathic behavior in a broad range of measures, not only in humans but also in other animals (see Christov-Moore et al., 2014 for a recent overview). Particularly relevant for the present studies are the difference regarding face processing. Women are more accurate and/or efficient in processing facial expressions of emotions in general (Hoffman, 1977; Hall and Matsumoto, 2004; Korb et al., 2015; Proverbio, 2017)(Hall, 1978; Hall and Matsumoto, 2004; Hoffman, 1977; Proverbio, 2017; Korb et al., 2015), and pain in particular (Keogh, 2014). Women show more facial mimicry than men (Dimberg and Lundquist, 1990) and are better than men in recognize pain also when it is expressed by the body posture (Walsh et al., 2017). In this section we therefore explored the effect of gender in our data. Given the low number of participants in each gender, these analyses should be interpreted in the light of their low statistical power. Detailed analyses are presented below, but in summary, we found no significant evidence for gender effects in our EEG study, and for the neuromodulation studies, we found no significant interactions between gender and neuromodulation.

Experiment 1: EEG study (15 males, 13 females)

Behaviouraly there was no evidence for gender differences in average donation or slopes: males average donation Hand Condition: M=2 SD=1.2, females average donation Hand Condition: M=2.3 SD=1.2, independent groups t-test t_(26)_=0.7 p=0.4; males average donation Face Condition: M=2 SD=1.3, females average donation Face Condition: M=2.2 SD=1.2, independent groups t-test t_(26)_=0.4 p=0.7; males average slope Hand Condition: M=0.5 SD=0.1, females average slope Hand Condition: M=0.4 SD=0.2, independent groups t-test t_(26)_=-1.2 p=0.2; males average slope Face Condition: M=0.5 SD=0.05, females average slope Face Condition: M=0.5 SD=0.07, independent groups t-test t_(26)_=-1.7 p=0.1.

To explore potential gender effects in the relationship between given donation and brain activity recorded in dSI-L and vSI-L, we first averaged together the slopes belonging to all the significant time-points in each task. Then using a t-test for independent groups for each cluster, we compared the average in the genders. Supplementary file 3 summarized the results. There is no evidence for gender difference in any of the clusters average.

Experiment 2: TMS study (9 males, 6 females)

We performed the same analysis on the slopes described in the main text adding ‘gender’ as a group factor. This new mixed ANOVA with 2 repeated factors (Condition and TMS) and a between participants factor (Gender) did not reveal any gender effect: main effect of Gender F_(1,13)_=0.008, p=0.9; Condition X Gender interaction F_(1,13)_=0.02, p=0.9; TMS X Gender interaction F_(1,13)_= 0.9, p=0.3; Condition X TMS X Gender interaction F_(1,13)_= 0.06, p=0.8.

Experiment 3: HD-tDCS study (13 males, 13 females)

As in experiment 2, we repeated the ANOVA described in the main text adding ‘Gender’ as a group factor. This new mixed ANOVA with 2 repeated factors (Stimulation and Condition) and a between participants factor (Gender) did not show any significant interaction with between Gender and Condition or Stimulation factors (Interaction Stimulation X Gender F_(1,23)=_0.3, p=0.6; Condition X Gender F_(1,23)=_1.5, p=0.2; Stimulation X Condition X Gender F_(1,23)_=0.03, p=1), suggesting that our the effect of our experimental manipulation does not depend on gender. However the ANOVA shows a main effect of Gender (F_(1,23)=_10, p=0.004): females had on average a steeper slope then males in all the tasks, independently of Condition and Stimulation. This result does not seem to be related to differences in empathic ability, since also in the control task females showed a steeper slope. It might be a general and unspecific difference in involvement in the task.

Figure 2C is not referred to. There is no Figure 4 but Figure 5 – which does not exist – is referred to. It appears that Figure 5 should be Figure 2C. Obviously fix and double check all figure references.

We now corrected the figure references.

Reviewer #2:

[…] 1) What do the results tell us about the nature of the activations in S1?I agree that approaches providing more causal insights are needed to advance the social neuroscience of affect and cognition. However, what remains to be shown in many cases, including my own research into this direction, is what the "causally involved" neural processes really mean in terms of "psychological/functional processes". This is why the third experiment using tDCS showing one possible mechanism – "perception of pain intensity" – is crucial, and it is currently way "undersold" in the paper/mentioned somewhat only in passing. I would incorporate this experiment into the write up of the paper from the beginning, including in the intro. And I would like to see a more extensive discussion of it and its implications. Moreover, this experiment tells us something about a possible shortcoming of the findings/an avenue for future research. The current approach basically only zooms in on activation in S1 in response to seeing hand movements in response to pain. It does unfortunately not exploit the opportunity – probably as this is much harder to implement due to the location of the brain areas involved – to also measure or inhibit activation in areas involved in face processing/facial affect discrimination. In an ideal world, the authors could aim for a double dissociation of areas involved in coding vicarious affect from observing hands vs. faces, and link this to differences in pain intensity decoding, and how they are related to prosocial action. This could be discussed more extensively as both a limitation and a future research avenue.

We have now incorporated the HD-tDCS experiment in the abstract and main manuscript.

In the Discussion section we now write: “Our results show that trial-by-trial amplitude of the EEG activity from ventral SI significantly explains changes in donation in the Face condition only. This effect could be driven by a covert internal simulation of the other’s facial expression or by overt facial mimicry. Interfering with facial mimicry has been shown to impair visual recognition of expressions (Oberman et al., 2007; Wood, et al., 2016) and interfering with activity in ventral somatosensory cortex alters emotion recognition from faces (Adolphs et al., 2000; Paracampo et al., 2017). Future research should neuro-modulate brain activity in ventral SI in addition to the hand representation we targeted here while measuring the willingness to help in order to further investigate the dissociation suggested by our results. The emergence of focused ultrasounds as a focal neuro-modulation method (Mueller et al., 2014; Lee et al., 2015; Lee et al., 2016), could enable such studies without the muscle artefacts inevitable with TMS. HD-tDCS, as used in our third experiment, also has the advantage not to cause muscle twitches, but lacks the focality to argue with confidence that one can disentangle the contribution of the face and hand region located only 2cm away.”

2) Power analysisI am truly surprised that the power analysis for the TMS experiment yielded a sample size of N=15, especially as the targeted power is unusually high with.95. It also doesn't sound very convincing that the effect size was based "on previous related findings", as the cited paper looked at TMS effects to decoding amusement from smiles. This sounds, sorry to be so direct, somewhat suspicious. It is also unclear for what kind of statistical tests the power analysis was computed? For regression analysis? t-tests?

We thank the reviewer for raising this point. We based our power analysis on the results of the work from Paracampo and colleagues (Paracampo, 2017) that also used online neuromodulation on the somatosensory cortex during an emotion recognition task and was characterized by a strong effect size (Cohen’s d = 0.94). That study was thus conducted by the same experimenter that was responsible for the TMS part in the current manuscript (RP); had a task that made similar cognitive demands; used an equivalent rTMS protocol and was targeting the same brain region (SI). As an additional analysis we now re-calculated the sample size including TMS studies in which participants are required to observe others and understand their behavior (Paracampo et al., 2017; Valchev et al., 2017; Paracampo et al., 2018; Tidoni et al., 2013). These studies are informative as they provide additional evidence on the magnitude of the expected effect produced by our TMS protocol but were not included in the present submitted manuscript as they did not satisfy all the criteria previously reported (or were not published when our manuscript was submitted). Pooling data from these experiments we obtained again a strong effect size (Cohen’s d = 0.89), we thus feel confident that the present sample size is appropriate.

In all the analyses power was computed for the specific comparison between participants’ performance in the active vs the sham condition (a t-test).

In the present version of the manuscript:

“Sample size for the TMS experiment was determined though a power analysis conducted using G*Power 3 (Faul et al. 2007), with power (1 – β) set at 0.95 and α = 0.05. The effect size was chosen based on the work of Paracampo and colleagues (Paracampo, 2017; Cohen’s d = 0.94), because it was conducted by the same experimenter that was responsible for the TMS part in the current manuscript (RP); had a task that made similar cognitive demands; used an equivalent rTMS protocol and was targeting the same brain region (SI). Comparable effect size (Cohen’s d = 0.89) was found when taking into account TMS studies in which participants are required to observe others and understand their behavior (Paracampo et al., 2017; Valchev et al., 2017; Paracampo et al., 2018; Tidoni et al., 2013). Because in our within subjects study we hypothesed a perturbation of the activity in SI, and therefore a reduction in performance as a consequence of the perturbation, we conducted a power analysis for the comparison between performance in active stimulation versus sham stimulation using a matched paired one-tailed t-test at the second (group) level. This analysis yielded a required sample size of 15 participants.”

Here, in Author response image 1, we report a screen shot of the performed analysis for the reviewer.

3) Face vs hand – sensitivity?The authors seem to suggest based on the TMS effects that the face condition does not causally involve S1 while the hand condition does. This is an unnecessary "binary" view on things, motivated by statistics showing effects only in one case and not the other. My take of the results would rather be that the effects for the hand are stronger, while it is unclear whether sensitivity (power? see above) of the study might not have been high enough for the face condition. I am saying this (also) because the face conditions showed, consistently, higher variability in donations. This might have driven the lack of detection effects also for this condition. I would thus tone down too "binary" interpretations of the TMS findings.

We thank the reviewer for rising this important point. As mentioned in reply to reviewer 1, that raised a similar point:

Our core result is an interaction between TMS condition (sham vs. active) and Stimulus condition (Face vs. Hand) on the regression slope. This provides evidence that the SI hand region is more necessary in determining donation based on information derived from the Hand than for the Face stimuli. We thus revised statements suggesting that we have evidence that the Hand region is not necessary in determining donation from the Face information to statements specifying that we have evidence that it is less involved.

Moreover, to address whether the non-significant effect of disrupting the SI hand region with TMS while viewing Face stimuli is due to a small sample size or is indicative of the absence of a sizable effect, we complement the more traditional frequentist analysis with a Bayesian statistic.

In subsection “Experiment2: TMS study” we write: “To test if the lack of effect in the Face condition was due to limited statistical power or provides evidence for the null hypothesis (H0) of no (sizable) effect of TMS, we used Bayesian statistics. We calculated an index of TMS effect for each participant in the Face condition as follows: slope in the Active session – slope in the Sham session. We then performed a Bayesian one sample T-Test using JASP (https://jasp-stats.org) that showed the null hypothesis is 5.7 times more likely than the alternative H1 (Bayes factor p(H0:index≥0|data)/p(H1:index<0|data)=5.7) providing positive evidence for the absence of a sizable effect in the Face condition (Kass and Raftery, 1995).”

In subsection “Experiment 3: HD-tDCS study”: “Since Hand and Face conditions showed opposite effect, we performed on the Color condition a 2-tails one sample Bayesian t-test. The Bayes factor indicated that the H0 was 3.1 times more likely than the H1, confirming that the Color condition does not change after HD-tDCS. The Hand and Face effects did not correlate with any of the tDCS side effect perceived by participants while performing the experiment (all p>0.05).”

As mentioned above, throughout the text we now focus on the interpretation derived from the significant interaction (I.e. the effect is smaller in the Face compared to the Hand condition), and now also address the possibility of a difference in sensitivity to TMS. In the Discussion section we now write:

“When pain is expressed by the reaction of the hand, the dorsal SI hand region activation correlates with decision making (EEG). Additionally, altering this activity changes (TMS) decision making, suggesting that the SI hand region activation feeds into the decision-making process. When pain is expressed by the face, the SI hand region activation correlates with decision making (EEG) but altering this activity influences decision making less. This latter finding is compatible with three interpretations. (i) Activity in the hand-region of SI is an epiphenomena, representing an imagination of what the painful stimulation would have felt on the hand (Fairhurst et al., 2012), that is not used in decision making. (ii) The SI hand region activity is used for decision making in the sham condition but can be substituted by alternative sources of information derived from the face elsewhere in the brain in the active TMS condition. Both these interpretations are reminiscent of the notion that pain information takes different paths based on the stimulus it is derived from (Keysers, Kaas and Gazzola, 2010; Lamm, Decety and Singer, 2011). (iii) Information in the SI hand region has a higher signal to noise ratio in the Face condition compared to the Hand condition, making it less susceptible to TMS interference.”

Reviewer #3:

[…] The link between empathy and prosocial behavior is not clear in this study. It has been suggested that empathic concern rather than personal distress contributes to prosocial behavior. This issue is not addressed in the study.

We agree that our study was not designed to address the interesting question of whether SI activity (which we measure and manipulate) is contributing to prosociality via an effect on empathic concern or personal distress. We now engage this issue in the Discussion section: “Furthermore we addressed the issue of how SI contributes to decision making, leveraging a third HD-tDCS experiment that allowed us to discriminate between perceptual or motivational contributions. Our results suggest that SI activation in the hand region contributes to prosocial decision-making by transforming the sight of hand-movements caused by a swat into a perception of pain-intensity, which then serves as an input to a decision-making process elsewhere. If this trial-by-trial perception is perturbed, our decision to help no longer optimally follows the trial-by-trial variance in pain experienced by others. This function is similar to the function that SI has during the observation of actions. For instance, disturbing the activity of the SI hand region with TMS makes ratings of the weight of an object seen lifted noisier compared to a sham condition, suggesting that the region is necessary for transforming observed hand kinematics into an estimate of the forces that have been acting on the hand (Valchev et al., 2017). A similar kinematic analysis may underpin the transformation of the observed hand kinematics following the swat into a painfulness estimate in our Hand condition. Affective social reactions, be they personal distress or empathic concern, would be informed by this kinematic analysis in SI, but require additional processes that the pain experience literature would ascribe to the anterior insula and cingulate (Lee and Tracey, 2010). In this interpretation, a neural network including SI informs the participant on how intense the swat was on a given trial and determines the ability of the participant to adjust the donation to the circumstances of a specific trial. In contrast, the mean donation could reflect trait differences in empathic concern (Davis, 1983) and money attitude (Yamauchi and Templer, 1982) (Supplemental analysis Correlation with self-reported questionnaires). A more in depth understanding of what emotional feelings (pain-like personal distress vs. more positively valenced empathic concern) accompany the motivational effect of SI activation on high pain trials remains unclear from our data and could be studied in future research by asking participants to provide specific ratings of their own affect on a trial-by-trial basis.” The Introduction has been also revisited in order to clarify the aims of our study and what aspects of empathy we focus on.

In the Supplementary material we write:

Correlation with self-reported questionnaires

At debriefing, participants completed the Interpersonal Reactivity Index questionnaire (Davis, 1983) and Money Attitude Scale (Yamauchi and Templer, 1982), to respectively measure empathy related traits and attitude towards money.

To assess if this personality traits explained participants’ differences in average donation we performed, in the costly helping experiment with the highest power (EEG, 28 participants) a multiple regression with the 4 IRI subscales (Empathic Concern, Perspective Taking, Fantasy Scale and Personal distress), and the MAS score as predictor of the average donation. In line with the notion that average donation reflects a trait, individual that gave high average donations in the Face condition also gave high average donations in the Hand condition (r(Hand,Face)=0.85). We therefore used the grand average donation across both conditions as the dependent measure in the regression. The regression resulted significant (F(5,22)=3.5, p=0.02, r2=0.45). In particular, Empathic Concern and the MAS significantly explained the average donation: the more participants rated themselves as having high empathic concern towards other and the less they affirmed to value money, the more they donated during the Costly Helping Paradigm (EC: b=0.4, p=0.02; MAS: b=-0.5, p=0.02; all others p>0.4).

Examining individual differences in the slope that links the intensity of pain in the InputMovie (as judged by an independent sample) with the trial-by-trial donation, lead to very different results. First, the slope in the Face and Hand condition were not strongly correlated (r(Hand,Face)=0.23), and multiple regressions performed separately for the Face and Hand slopes revealed that none of the scales significantly explained variance in the slopes (all p>0.1).

The above results are not replicated within the smaller group of participants for the TMS study (N=15) and should therefore be considered tentative.

The role of the SI in prosocial behavior is not properly justified in the introduction.There are no clear predictions in the presented in the Introduction.The role of the face condition as presented in the Introduction is not clear

We revised the Introduction extensively and hope this clarifies these issues. In particular, with regard to the predictions, we now write: “In a first experiment, we investigate whether activation of the hand region of the left SI, as measured with EEG, explains prosocial behavior. The SI hand region was identified in an independent pool of participants by correlating fMRI BOLD responses within SI with subjective experience of pain elicited by electrical stimulations on participant’s right hand. We hypothesized that activation of the hand region of SI would correlate with the donation given by the participants in the Hand condition, when the intensity of the stimulation had to be deduced from the hand movement. In the Face condition we predicted that activity more ventrally in SI, where facial expressions are represented, would correlate with the donation, as the relevance of facial mimicry has been highlighted in many studies (Oberman et al., 2007; Wood et al., 2016; Hess and Fischer, 2013; Fischer and Hess, 2017), and ventral somatosensory cortex causally contributes to emotion perception from facial expression (Adolphs et al., 2000; Paracampo et al., 2017) and recognizing emotions from visually presented facial expressions requires ventral somatosensory-related cortices (Adolphs et al., 2000). As mentioned above, for the hand region of SI during the Face condition, we had less defined predictions: the presence or absence of correlation of SI hand region activity with the donation while perceiving facial expressions will inform whether facially deduced pain intensity is re-represented in the SI locations reflecting the inferred origin of that pain.

In a second experiment, we then perturbed the SI activity of the hand region with repetitive TMS (rTMS) to test whether disturbing SI vicarious activity altered prosocial behavior. Because disrupting SI activity using TMS or neurological lesions has been shown to alter the accuracy with which participants perceive some emotions (Adolphs et al., 2000; Paracampo et al., 2017) and hand actions (Valchev et al., 2017), and because in the nociceptive literature, SI has been associated more with perceptual than motivational processes (Keysers et al., 2010; Lee and Tracey 2010) we expected TMS over the hand-region of SI to disrupt the accuracy with which participants can transform the observed kinematics of the belt and hand into an accurate feeling for how painful this particular stimulation was for the other. We thus expect decision-making to become noisier, and less attuned to the level of pain experienced by the other on a trial-by-trial basis particularly when the reaction of the hand is the only source of information for the decision-making (Hand condition). This effect would be weaker or absent when information is derived from the Face, where alternative sources of information are available.

Finally, we used data from a third experiment to explore whether a disruption of the perception of pain intensity indeed mediates how disrupting SI activity alters decision-making. Brain activity in SI was altered using high-definition tDCS while participants had to rate how much pain the person in the Hand and Face movies experienced on a trial by trial basis. Because the specific montage used in this experiment was expected to facilitate brain activity under the anode placed over the SI hand region and inhibit brain activity under the return cathodes, one of which was placed over the face region of SI, we expected the accuracy of the ratings to increase in the Hand stimuli and decrease in the Face stimuli.”

Materials and methods section:It is not clear why the authors focused on the left SI and not the right. Other studies suggest that the right S1 is involved in empathy and there is also evidence for bilateral involvement.

From the current text: “We focused on the left hemisphere, because electrical stimulation was always delivered to the right hand of both the confederate shown in the movies, and of the participants in the pain localizer.”

“There is evidence for a bilateral receptive field in the Brodmann 1 and 2 sub-regions of SI (Iwamura et al., 2002), and for the involvement of the right hemisphere in the perception of emotion (not including pain) from facial expressions (Adolphs et al., 2000; bilateral activation are reported in Ashar et al., 2017; Lamm et al., 2011; Cui et al., 2015; Carr et al., 2003), and hands movement (Christov-Moore and Iacoboni, 2016). Figure 3D shows the signal originating from mirroring our left ROIs to the right hemisphere, and in yellow the time points significantly explaining the donation. For the hand region of SI (d-SI), results for the two hemispheres are very similar suggesting a lack of clear hemispheric specificity (Figure 3D and Supplementary Information). For the more ventral, putative face region of SI (v-SI), responses appear stronger on the right hemisphere, in line with previous findings (Adolphs et al., 2000).”

“Our EEG findings therefore suggest that while witnessing the pain of another person, the magnitude of brain activity in the hand region of SI (d-SI) could inform decision making. To examine its causal contribution to decision making, in a second experiment we use TMS to disturb the activity of the SI hand region. We will target the left hemisphere because it is contralateral to the hand that is stimulated in the confederate.”

TMS experimentIsn't it problematic to perform the task twice?

Our initial piloting revealed that testing participants over multiple days led to increased scepticism about the cover story. For this reason, we decided (a) to use different pools of participants for each experiment (EEG vs TMS vs HD-tDCS), and (b) to limit the number of trials in our TMS study to what could fit a within subject design. Participants came once to the lab to perform the TMS task and had both Active and Sham TMS stimulation in a randomized fashion. None of the participant ever performed the task twice, but the real and sham session were presented as a single experiment. The original pool of selected videos was divided into two blocks for each condition, and each block was then split in two subgroups, which were then randomly attributed to the sham and real TMS sessions. Block attribution was then randomized between participants. The sham and real TMS sessions did never contain the same videos (i.e. each session contained a different subsample from the original pool of videos), but some videos might have been shown twice within each block. We tried to clarify this in the Materials and methods section.

In the Costly Helping Visual Stimuli section: “Because the intensity of the OutputMovie depended on the participant’s donation it was impossible to precisely predict the number of videos needed for each intensity and participant. This means that in some cases, the number of recorded videos was lower than the number of actual presentation of a particular intensity, and few videos had to be shown more than once. Care was taken to maximize the distance between repetitions of the same stimulus. The number of repeated videos is low, and it was never the case that a participant saw all the movies twice.”

In the TMS study section: “The conditions were presented in four blocks (two blocks for the Face and two for the Hand condition), separated by a long break in the middle in which participants further interacted with the confederate. Each block was equally divided in two parts which were assigned to active and sham TMS in a pseudo-randomized order (e.g. for one participant: ActiveTMSFaceBlock1part1, ShamTMSFaceBlock1part2, ShamHandBlock1part2, ActiveHandBlock1part2, long break, ActiveHandBlock2part1, ShamHandBlock2part2, ActiveFaceBlock2part2, ShamFaceBlock2part1). While some videos might have been shown twice within block1 or block2, video in block1 were different from videos in block2.”

Additionally, during the debriefing, we asked our participants to indicate on a 7 step scales (from strongly disagree to strongly agree) how well the following statement applied to them: “You think you saw twice the same movie.” In average, participants of the EEG experiment (in which a number of InputMovie was also shown twice) reported a value of 6.1 (see Table and Figure 1—figure supplement 1), and those of the TMS one of 6.3, suggesting seeing the movies twice was not easily recognized by the our participants. Interestingly, when running a correlation between the responses to the statement “You think you saw twice the same movie” and those to the statement “You think the experimental setup was realistic enough to believe it”, the correlations become significant when the participants originally exclude because had doubts regarding the experiment were included in these correlations, indicating a possible effect of seeing the movie twice which was taken care off by our exclusion criteria.

Average (Quartiles Q1, Q2 and Q3) for: “You think you saw twice the same movie”Average (Quartiles Q1, Q2 and Q3) for: “You think the experimental setup was realistic enough to believe it”r(You think you saw twice the same movie”, “You think the experimental setup was realistic enough to believe it”), p valueEEG all participants3.8 (4, 3.25)5.8 (6, 1)r(35)=-0.45 p=.009EEG after participant’s exclusion3.5 (3, 3)6.1 (6, 1)r(28)=-0.32 p=.09TMS all participants3.4 (2.5, 3.75)5.5 (6, 1.75)r(18)=-0.47 p=.049TMS after participant’s exclusion3.0 (2, 2.5)6.3 (6, 1)r(15)=-0.06 p=.8

In subsection “Costly Helping Visual Stimuli” we now only mention: “During debriefing, we additionally asked our participants to indicate on a 7-step scale (from 1=strongly disagree to 7=strongly agree) how well the following statement applied to them: “You think you saw twice the same movie.” In average, participants of the EEG experiment reported a value of 6.1 (Figure 1—figure supplement 1), and those of the TMS one of 6.3, suggesting seeing the movies twice was not easily recognized by our participants.”

It is important to examine the effect of order of the stimulation.How accurate was the location of S1 and why no navigation method was applied?

We thank the reviewer for raising this point allowing us to be more precise on the technique used to locate our target region in the TMS experiment. The anatomo-functional approach we use has been used in several neuromodulation studies showing high precision and reliability (Avenanti et al., 2017; Jacquet and Avenanti, 2015; Harris et al., 2002; Merabet et al., 2004; Fiorio and Haggard, 2005; Azañón and Haggard, 2009; Balslev, 2004; Valchev et al., 2017; Tegenthoff et al., 2005). Based on that previous literature that also tested the accuracy in the localization of the hand representation in S1 with neuroimaging tools, we are quite confident that the coil was positioned in a position that has previously been shown to interfere with that specific hand representation. In the manuscript we now write: “A large body of evidence shows that the hand area in the somatosensory cortex can be successfully targeted positioning the coil 1–4cm posterior to the motor hotspot (Avenanti et al., 2017; Jacquet and Avenanti, 2015; Harris et al., 2002; Merabet et al., 2004; Fiorio and Haggard, 2005; Azañón and Haggard, 2009; Balslev, 2004; Valchev et al., 2017; Tegenthoff et al., 2005). This approach is based on the close correspondence between the motor and somatic homunculi (Buccino et al., 2001; Yang et al., 1994; Schulz et al., 2004; Nakamura et al. 1998; Amunts and Zilles, 2015; Kuehn et al., 2014). In line with this, we identified our region of interest using a two-step procedure. First, we localized the hand region in the primary motor cortex, corresponding to the optimal scalp position (OSP) for evoking MEPs in the FDI muscle. After that, we moved the coil 2cm backward following a parasagittal plane assuming that this displacement would not produce effects on M1. We tested this assumption directly by checking that TMS pulses applied at 105% rMT with the coil in the final target position did not elicit any detectable MEPs”.

It seems that testing if TMS interferes with the relationship between intensity and donation is post-hoc. This analysis is not discussed in the Introduction and not justified.

We now clarify this prediction in the Introduction: “In a second experiment, we then perturbed the SI activity of the hand region with repetitive TMS (rTMS) to test whether disturbing SI vicarious activity altered prosocial behavior. Because disrupting SI activity using TMS or neurological lesions has been shown to alter the accuracy with which participants perceive some emotions (Adolphs et al., 2000; Paracampo et al., 2017) and hand actions (Valchev et al., 2017), and because in the nociceptive literature, SI has been associated more with perceptual than motivational processes (Keysers et al., 2010; Lee and Tracey, 2010) we expected TMS over the hand-region of SI to disrupt the accuracy with which participants can transform the observed kinematics of the belt and hand into an accurate feeling for how painful this particular stimulation was for the other. We thus expect decision-making to become noisier, and less attuned to the level of pain experienced by the other on a trial-by-trial basis particularly when the reaction of the hand is the only source of information for the decision-making (Hand condition). This effect would be weaker or absent when information is derived from the Face, where alternative sources of information are available”.

Reviewer #4:[…] There are some statements made in the intro that I think are oversimplifications that could be improved. I think the existing knowledge about prosocial behavior, about the role of empathy in altruism, and about existing neuroscience tools besides TMS to study this problem are underestimated. There is abundant evidence that empathy can promote altruism but also that you can get helping for other reasons (display, status, self-interest) and that sometimes people feel empathy but don't help (because of fear, conflict of interest, distress etc.). There are also ways that neuroscience can and already has addressed some of these issues. The addition of TMS is nice but it is not necessary to make it seem like the whole field to this point has been a morass of confusion in order to justify this work. The very need to shore up some issues with our understanding of the role of this region is plenty of motivation to perform and publish this work.

We edited the entire manuscript to provide more pointers towards non-TMS literature that is relevant to the link between empathy and prosociality, to avoid suggesting that TMS is the only way to address the role of empathy in altruism, and to focus more on our specific question: does a somatosensory simulation of the observed reactions to noxious stimulations inform prosocial decision making, and do perceptual changes mediate such effect?

We felt that providing a more extensive review of the link between empathy and prosociality (e.g. empathy induction techniques), would bring the attention away from the somatosensory simulation focus of the experiment.

Note that there is a major mis-step in the Discussion section where it says multiple times that because the face TMS did not alter donations that means that SI activation does not influence decision making. This is incorrect. What you have shown is that it is not necessary for or fully explanatory of decision making in the current context. There are multiple reasons to not make the assumption that you did. For example, people are expert facial readers who can make explicit and top-down inferences about internal states from expressions without having to rely on neural/physiological feeling states once they are "trained up" in an embodied system and learn cultural display rules. In addition, people can use that information normally as inputs to the decision even if they can find other ways (like the aforementioned) to make the choice when that input is not available.

This issue was recurrent through the other reviewer’s core comments as well, and we also totally agree. We therefore revised the entire manuscript to down-tone those statements. In the Discussion section we additionally write: “When pain is expressed by the face, the SI hand region activation correlates with decision making (EEG) but altering this activity influences decision making less. This latter finding is compatible with three interpretations. (i) Activity in the hand-region of SI is an epiphenomenon, representing an imagination of what the painful stimulation would have felt on the hand (Fairhurst et al., 2012), that is not used in decision making. (ii) The SI hand region activity is used for decision making even for the Face condition in the sham condition but can be substituted by alternative sources of information derived from the face elsewhere in the brain in the active TMS condition. Both these interpretations are reminiscent of the notion that pain information takes different paths based on the stimulus it is derived from (Keysers et al., 2010; Lamm et al., 2011). (iii) Information in the SI hand region has a higher signal to noise ratio in the Face condition compared to the Hand condition, making it less susceptible to TMS interference. Differentiating these alternatives will require further experimentation”.

I am concerned about the distribution of pain intensity in Table 2. The TMS face condition has a skewed distribution toward the lowest intensity and does not have the second peak of higher intensity that is available for the other three conditions. I think you have to assume that any differences in the Face condition, especially null effects in the TMS study could reflect small(er) effect sizes because there is not enough high intensity information to drive effects.

The reviewer refers to this particular table:

Perceived IntensityEEG expTMS exp/per sessionHandFaceHandFace21323410323104404445271010661112671Average Intensity4 ± 1.43.7 ± 1.73.9 ± 1.23.8 ± 1.6

Table 2: Number of videos for each intensity presented as InputMovie in the EEG and TMS experiment. Last line calculates the average movie intensity and its standard deviation presented for each condition and experiment.

During the stimuli validation, we had classified stimuli in three categories: low intensity (perceived intensity 2 and 3), medium intensity (perceived intensity 4) and high intensity (perceived intensity 5 and 6). For both TMS tasks, we had 14 trials of low, 4 of medium and 12 of high perceived intensity, chosen to keep the number of medium stimuli (close to the average) low to maximize the power of the regression analysis. In an analysis we do not report in the manuscript we then averaged the donation given by the participants in the high, medium and low intensity trials for each task (Face or Hand) and session (Active vs Sham TMS), then conducted a repeated measure ANOVA with 3 factors (Task, TMS, Intensity). The ANOVA showed that the TMS effect (Donation Active – Donation Sham) depended on Intensity more for the Hand than for the Face (triple interaction (F_(1,14)_=4.7, p=0.047, Figure below). Post-hoc analyses reveals that the TMS effect (Active – Sham donation) was significant only for the low intensity Hand stimuli: during active TMS stimulation over SI participants tended to donate more than during sham stimulation when watching hand video depicting low intensity, as if they failed to detect that the confederate is in so little pain, that they needn’t give up some of their money.

**Author response image 2. respfig2:** 

Only later did we transit to a more metric analysis, as presented in the paper, and we agree that this creates this unfortunate incongruity in distribution. However, the power of the regression (which is proportional by the standard deviation of the regressor) is higher for the Face condition, in which we did not find a significant TMS effect, than for the Hand condition, in which we did find a significant and significantly larger TMS effect. We would thus expect that if differences in power were to bias a result, it would bias it in the direction opposite to the finding we found.

From a statistical point of view, we hope that the categorical ANOVA helps mitigate some of the reviewer’s concern. However, this does not mitigate the concern that SI representations to more extreme observed reactions (Face) might be harder to disrupt than those to more subtle reactions (Hand), and now also directly suggest that higher signal to noise in the Face stimuli (due to the more extreme stimuli) may have accounted for the smaller effect of TMS as option (iii) in the discussion already quoted above “When pain is expressed by the face, the SI hand region activation correlates with decision making (EEG), but altering this activity influences decision making less. This latter finding is compatible with three interpretations. … (iii) Information in the SI hand region has a higher signal to noise ratio in the Face condition compared to the Hand condition, making it less susceptible to TMS interference. Differentiating these alternatives will require further experimentation”.

Note that the Preston and de Waal, 2002 paper is cited at the beginning as evidence that there is a lot of confusion over empathy producing altruism. That is not an accurate depiction of the contents of that paper. It is also not included in the reference section. The fact that the 2002 paper was the first systematic review to even suggest that empathy relied on overlapping neural substrates between felt and observed pain is overlooked and should be cited. That theory was even based on evidence from Adolphs and colleagues showing that somatosensory cortex per se is needed to decode facial emotion--a study that used lesion patients (and, thus, has good causal inference power, in conflict with claims here that this is the first time ever that one could make claims about causality with this region in emotion perception or empathy…). I think this citation issue and credit needs to be addressed. You can also see a new review, de Waal and Preston, 2017, for an updated version that clearly includes somatosensory in the map of empathy-related areas.

We apologize for the confusion, and now changed the references accordingly: “In this perspective the emotional states of others are understood through personal, embodied representations that allow empathy and accuracy in perceiving other emotions to increase based on the observer’s past experiences (de Waal and Preston, 2017; Preston et al., 2002).” And later regarding to somatosensory cortices “and consequently included in the network of regions participating in human empathy (de Waal and Preston 2017; Keysers et al., 2010)”

I'm not sure you can conclude that SI activity predicting the donation in the second half means that only the reaction to the pain is relevant since the response to both (belt hitting hand and person reacting) could be used and have to summate or build with additional processing to reach the threshold necessary to explain donations. Also, the belt video I saw does not have a very fast, salient, painful looking hit, so I would assume that the NS for that segment can easily be an effect size problem with the stimuli that doesn't look super painful.

We fully agree with the reviewer. In the original manuscript, we meant to separate whether SI was representing the actions of the person wielding the belt (seen mainly in the first half of the movie) vs. what happens to the hand of the victim (in the second half of the movie). We did not mean to suggest that the deformation of the hand caused by the belt was not informative. To clarify this distinction, we reformulated the text as follows: “In the Hand movie, movements are displayed both in the first half of the video by the agent wielding the belt and in the second half by the victim’s hand being compressed by and reacting to the swat. Activity in SI has been shown to potentially encode all of these (Keysers et al., 2010). Interestingly, SI activity significantly predicted the donation only during the second half, in which the victim’s hand is compressed by the belt and reacts to it. This suggests that it is SI’s ability to represent the impact of the belt on the hand or the reaction of the victim to the stimulation on the hand that induced the prosocial decision making.”

To further assess if the first half of the Hand movies contain information about the perceived intensity of the stimulation (conveyed by the agent moving the belt) we cut each Hand videos used in the HD-tDCS experiment in two halves. The first half started at the beginning for the video and lasting 900ms and showed the agent wielding the belt. The second half started with the belt hitting the hand and lasted until the end of the video, and thus showed the belt compressing the hand and the reaction of the hand.

A total of 41 participants took part in this experiment. They were divided in two groups: one group, composed by 20 of them (12 females, mean age of 28.6 DS=8.5) watched only the first half of all the movies, the other 21 (12 females, mean age of 27 DS=2.9) watched only the second half of each clip. Participants performed the experiment online (Lime survey platform). Participants received the instruction to rate from 1 to 8 how much pain the person in the video will be or is experiencing, respectively for the FirstHalf group and SecondHalf group.

To assess the relationship between the ratings given when watching one of the two halves of the movies and the ratings given when watching the entire Hand movies, for each participant we calculated the correlation coefficient between the rating given and the perceived intensity assigned in the validation procedure describe in the Method section, HD-tDCS experiment. Some of the subjects viewing the FirstHalf movies rated all movies with the minimum perceived intensity possible. Because the lack of variance in their ratings would not allow to calculate the correlation, we approximated the rating of a trial randomly chosen of 0.1. This small trick allowed us to calculate the correlation coefficient with a minimal deformation of the data. A similar procedure is routinely used to calculate of signal detection theory measures (e.g. Stanislaw and Todorov, 1999). We transformed the correlation coefficient using the Fisher transformation before applying parametric statistics to the transformed values.

At the group level we analysed the distribution of the normalized correlation coefficients: if the clips did not carry systematic information about the intensity of the stimulation, the correlation coefficients would be randomly distributed around zero. This is what we found for the FirstHalf group (M=0.05, SD=0.2, t(19)=1.04, p=0.3). In contrast, the SecondHalf group showed a robust correspondence with the ratings of the validation group (M=0.6, SD=0.2, t(20)=11, p<0.0001).

We have so far not included these results in the manuscript but would be happy to do so if the reviewer or editor feels this would be appropriate.